# Dynamic Graph Representation Learning via Graph Transformer Networks

## Abstract

Dynamic graph representation learning is an important task with widespread applications. Previous methods on dynamic graph learning are usually sensitive to noisy graph information such as missing or spurious connections, which can yield degenerated performance and generalization. To overcome this challenge, we propose a Transformer-based dynamic graph learning method named Dynamic Graph Transformer (DGT) with *spatial-temporal encoding* to effectively learn graph topology and capture implicit links. To improve the generalization ability, we introduce two complementary *self-supervised pre-training tasks* and show that jointly optimizing the two pre-training tasks results in a smaller Bayesian error rate via an information-theoretic analysis. We also propose a *temporal-union graph* structure and a *target-context node sampling strategy* for an efficient and scalable training. Extensive experiments on real-world datasets illustrate that DGT presents superior performance compared with several state-of-the-art baselines.

## 1 Introduction

In recent years, graph representation learning has been recognized as a fundamental learning problem and has received much attention due to its widespread use in various domains, including social network analysis (Kipf & Welling, 2017; Hamilton et al., 2017), traffic prediction (Cui et al., 2019; Rahimi et al., 2018), knowledge graphs (Wang et al., 2019a;b), drug discovery (Do et al., 2019; Duvenaud et al., 2015), and recommendation systems (Berg et al., 2017; Ying et al., 2018). Most existing graph representation learning work focuses on static graphs. However, real-world graphs are intrinsically dynamic where nodes and edges can appear and disappear over time. This dynamic nature of real-world graphs motivates dynamic graph representation learning methods that can model temporal evolutionary patterns and accurately predict node properties and future edges.

Recently, several attempts (Sankar et al., 2018; Pareja et al., 2020; Goyal et al., 2018) have been made to generalize graph learning algorithms from static graphs to dynamic graphs by first learning node representations on each static graph snapshot then aggregating these representations from the temporal dimension. However, these methods are vulnerable to noisy information such as missing or spurious links. This is due to the ineffective message aggregation over unrelated neighbors from noisy connections. The temporal aggregation makes this issue severe by further carrying the noise information over time. Over-relying on graph structures makes the model sensitive to noisy input and can significantly affect downstream task accuracy. A remedy is to consider the input graph as fully connected and learn a graph topology by assigning lower weights to task-irrelevant edges during training (Devlin et al., 2019). However, completely ignoring the graph structure makes the optimization inefficient because the model has to estimate the underlying graph structure while learn model parameters at the same time. To resolve the above challenges, we propose a Transformer-based dynamic graph learning method named Dynamic Graph Transformer (DGT) that can "*leverage underlying graph structures*" and "*capture implicit edge connections*" to balance this trade-off.

Transformers (Vaswani et al., 2017), designed to automatically capture the inter-dependencies between tokens in a sequence, have been successfully applied in several domains such as Natural Language Processing (Devlin et al., 2019; Brown et al., 2020) and Computer Vision (Dosovitskiy et al., 2020; Liu et al., 2021). We summarize the success of Transformers into three main factors, which can also help resolve the aforementioned challenges in dynamic graph representation learning: (1) **fully-connected self-attention**: by modeling all pair-wise node relations, DGT can capture implicit edge connections,

thus become robust to graphs with noisy information such as missing links; (2) ***positional encoding***: by generalizing positional encoding to the graph domain using spatial-temporal encoding, we can inject both spatial and temporal graph evolutionary information as inductive biases into DGT to learn a graph's evolutionary patterns over time; (3) ***self-supervised pre-training***: by optimizing two complementary pre-training tasks, DGT presents a better performance on the downstream tasks.

Though powerful, training Transformers on large-scale graphs is non-trivial due to the quadratic complexity of the fully connected self-attention on the graph size (Zaheer et al., 2020; Wang et al., 2020). This issue is more severe on dynamic graphs as the computation cost grows with the number of time-steps in a dynamic graph (Pareja et al., 2020; Sankar et al., 2018). To make the training scalable and independent of both the graph size and the number of time-steps, we first propose a *temporal-union graph* structure that aggregates graph information from multiple time-steps into a unified meta-graph; we then develop a two-tower architecture with a novel *target-context node* sampling strategy to model a subset of nodes with their contextual information. These approaches improve DGT's training efficiency and scalability from both the temporal and spatial perspectives.

To this end, we summarize our contributions as follows: (1) a two-tower Transformer-based method named DGT with spatial-temporal encoding that can capture implicit edge connections in addition to the input graph topology; (2) a *temporal-union graph* data structure that efficiently summarizes the spatial-temporal information of dynamic graphs and a novel *target-context node* sampling strategy for large-scale training; (3) two complementary pre-training tasks that can facilitate performing downstream tasks and are proven beneficial using information theory; and (4) a comprehensive evaluation on real-world datasets with ablation studies to validate the effectiveness of DGT.

## 2 PRELIMINARIES AND RELATED WORKS

In this section, we first define dynamic graphs, then review related literature on dynamic graph representation learning and Transformers on graphs.

**Dynamic graph definition.** The nodes and edges in a dynamic graph may appear and disappear over time. In this paper, we define a dynamic graph as a sequence of static graph snapshots with a temporal order $\mathbb{G} \triangleq \{\mathcal{G}_1, \ldots, \mathcal{G}_T\}$, where the $t$-th snapshot graph $\mathcal{G}_t(\mathcal{V}, \mathcal{E}_t)$ is an undirected graph with a shared node set $\mathcal{V}$ of all time steps and an edge set $\mathcal{E}_t$. We also denote its adjacency matrix as $\mathbf{A}_t$. Our goal is to learn a latent representation of each node at each time-step $t$, such that the learned representation can be used for any specific downstream task such as link prediction or node classification. Please notice that the shared node set $\mathcal{V}$ is not static and will be updated when new snapshot graph arrives, which is the same as Sankar et al. (2018); Pareja et al. (2020).

**Dynamic graph representation learning.** Previous dynamic graph representation learning methods usually extend static graph algorithms by further taking the temporal information into consideration. They can mainly be classified into three categories: (1) *smoothness-based methods* learn a graph autoencoder to generate node embeddings on each graph snapshot and ensure the temporal smoothness of the node embeddings across consecutive time-steps. For example, DYGEM (Goyal et al., 2018) uses the learned embeddings from the previous time-step to initialize the embeddings in the next time-step. DYNAERNN applies RNN to smooth node embeddings at different time-steps; (2) *Recurrent-based methods* capture the temporal dependency using RNN. For example, GCRN (Seo et al., 2018) first computes node embeddings on each snapshot using GCN (Defferrard et al., 2016), then feeds the node embeddings into an RNN to learn their temporal dependency. EVOLVEGCN (Pareja et al., 2020) uses RNN to estimate the GCN weight parameters at different time-steps; (3) *Attention-based methods* use self-attention mechanism for both spatial and temporal message aggregation. For example, DYSAT (Sankar et al., 2018) propose to use the self-attention mechanism for temporal and spatial information aggregation. TGAT (Xu et al., 2020) encodes the temporal information into the node feature, then applies self-attention on the temporal augmented node features. However, *smoothness-based methods* heavily rely on the temporal smoothness and are inadequate when nodes exhibit vastly different evolutionary behaviors, *recurrent-based methods* scale poorly when the number of time-steps increases due to the recurrent nature of RNN, *attention-based methods* only consider the self-attention on existing edges and are sensitive to noisy graphs. In contrast, DGT leverages Transformer to capture the spatial-temporal dependency between all nodes pairs, does not over-relying on the given graph structures, and is less sensitive to noisy edges.

**Graph Transformers.** Recently, several attempts have been made to leverage Transformer for

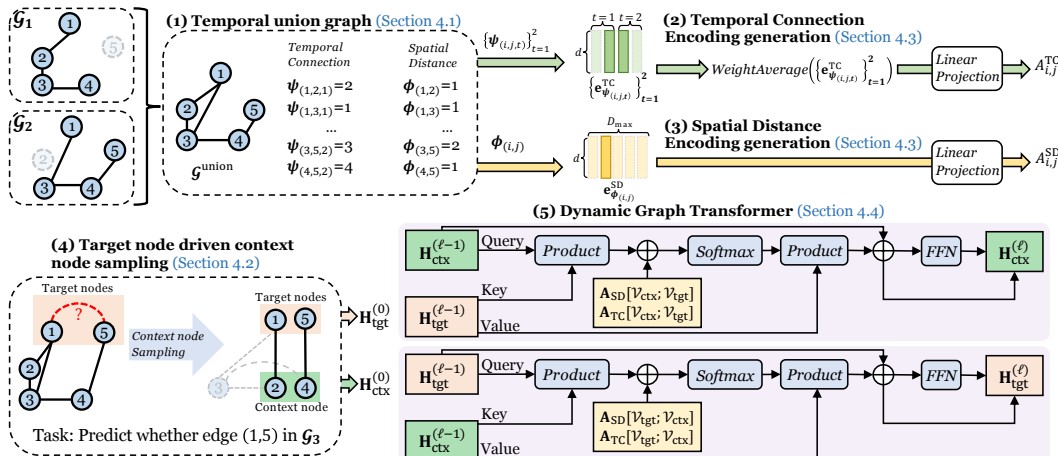

Figure 1: Overview of using DGT for link prediction. Given snapshot graphs $\{\mathcal{G}_1, \mathcal{G}_2\}$ as input, **(1)** we first generate the temporal union graph with the considered max shortest path distance $D_{\max} = 5$, and its associated **(2)** temporal connection encoding and **(3)** spatial distance encoding. Then, the encodings are mapped into $A_{i,j}^{\text{TC}}, A_{i,j}^{\text{SD}}$ for each node pairs $(i, j)$ using a fully connected layer. To predict whether an edge exists in future graph $\mathcal{G}_3$, we first **(4)** sample target nodes and context nodes, and then apply **(5)** DGT to encode target nodes and context nodes separately.

graph representation learning. For example, GRAPHORMER (Ying et al., 2021) and GRAPHTRANS-FORMER (Dwivedi & Bresson, 2020) use scaled dot-product attention (Vaswani et al., 2017) for message aggregation and generalizes the idea of positional encoding to graph domains. GRAPH-BERT (Zhang et al., 2020) first samples an egocentric network for each node, then order all nodes into a sequence based on node importance, and feed into the Transformer. However, GRAPHORMER is only feasible to small molecule graphs and cannot scale to large graphs due to the significant computation cost of full attention; GRAPHTRANSFORMER only considers the first-hop neighbor aggregation, which makes it sensitive to noisy graphs; GRAPHBERT does not leverage the graph topology and can perform poorly when graph topology is important. In contrast, DGT encodes the input graph structures as an inductive bias to guide the full-attention optimization, which balances the trade-offs between noisy input robustness and efficiently learning an underlying graph structure. A detailed comparison is deferred to Appendix D.

## 3 METHOD

In this section, we first introduce the temporal union-graph (in Section 3.1) and our sampling strategy (in Section 3.2) that can reduce the overall complexity from the temporal and spatial perspectives respectively. Then, we introduce our spatial-temporal encoding technique (in Section 3.3), describe the two-tower transformer architecture design, and explain how to integrate the spatial-temporal encoding to DGT (in Section 3.4). Figure 1 illustrates the overall DGT design.

### 3.1 TEMPORAL-UNION GRAPH GENERATION

One major challenge of applying Transformers on graph representation learning is its significant computation and memory overhead. In Transformers, the computation cost of self-attention is $\mathcal{O}(|\mathcal{E}| d)$ and its memory cost is $\mathcal{O}(|\mathcal{E}| + |\mathcal{V}| d)$. When using full attention, the computation graph is fully connected with $|\mathcal{E}| = |\mathcal{V}|^2$, where the overall complexity is quadratic in the graph size. On dynamic graphs, this problem can be even more severe if one naively extends the static graph algorithm to a dynamic graph, e.g., first extracting the spatial information of each snapshot graph separately, then jointly reasoning the temporal information on all snapshot graphs (Sankar et al., 2018; Pareja et al., 2020). By doing so, the overall complexity grows linearly with the number of time-steps $T$, i.e., with $\mathcal{O}(|\mathcal{V}|^2 T d)$ computation and $\mathcal{O}(|\mathcal{V}|^2 T + |\mathcal{V}| T d)$ memory cost. To reduce the dependency of the overall complexity on the number of time-steps, we propose to first aggregate dynamic graphs $\mathbb{G} = \{\mathcal{G}_1, \ldots, \mathcal{G}_T\}$ into a *temporal-union graph* $\mathcal{G}^{\text{union}}(\mathcal{V}, \mathcal{E}')$ then employ DGT on the generated temporal-union graph, where $\mathcal{E}' = Unique\{(i, j) : (i, j) \in \mathcal{E}_t, \ t \in [T]\}$ is the set of

all possible unique edges in $\mathbb{G}$. As a result, the overall complexity of DGT does not grow with the number of time-steps. Details on how to leverage spatial-temporal encoding to recover the temporal information of edges are described in Section 3.3.

## 3.2 TARGET NODE DRIVEN CONTEXT NODE SAMPLING

Although the temporal-union graph can alleviate the computation burden from the temporal dimension, due to the overall quadratic complexity of self-attention with respect to the input graph size, scaling the training of Transformer to real-world graphs is still non-trivial. Therefore, a properly designed sampling strategy that makes the overall complexity independent with graph sizes is necessary. Our goal is to design a sub-graph sampling strategy that ensures a fixed number of well-connected nodes and a lower computational complexity. To this end, we propose to first sample a subset of nodes that we are interested in as *target nodes*, then sample their common neighbors as *context nodes*.

Let *target nodes* $\mathcal{V}_{\text{tgt}} \subseteq \mathcal{V}$ be the set of nodes that we are interested in and want to compute its node representation. For example, for the link prediction task, $\mathcal{V}_{\text{tgt}}$ are the set of nodes that we aim to predict whether they are connected. Then, the *context nodes* $\mathcal{V}_{\text{ctx}} \subseteq \{\mathcal{N}(i) \mid \forall i \in \mathcal{V}_{\text{tgt}}\}$ are sampled as the common neighbors of the target nodes. Notice that since context nodes $\mathcal{V}_{\text{ctx}}$ are sampled as the common neighbors of the target nodes, they can provide local structure information for nodes in the target node set. Besides, since two different nodes in the target node set can be far apart with a disconnected neighborhood, the neighborhood of two nodes can provide an approximation of the global view of the full graph. During the sampling process, to control the randomness involved in the sampling process, $\mathcal{V}_{\text{ctx}}$ are chosen as the subset of nodes with the top-$K$ joint Personalized PageRank (PPR) score (Andersen et al., 2006) to nodes in $\mathcal{V}_{\text{tgt}}$, where PPR score is a node proximity measure that captures the importance of two nodes in the graph. More specifically, our joint PPR sampler proceeds as follows: First, we compute the approximated PPR vector $\boldsymbol{\pi}(i) \in \mathbb{R}^N$ for all node $i \in \mathcal{V}_{\text{tgt}}$, where the $j$-th element in $\boldsymbol{\pi}(i)$ can be interpreted as the probability of a random walk to start at node $i$ and end at node $j$. We then compute the approximated joint PPR vector $\hat{\boldsymbol{\pi}}(\mathcal{V}_{\text{tgt}}) = \sum_{i \in \mathcal{V}_{\text{tgt}}} \boldsymbol{\pi}(i) \in \mathbb{R}^N$. Finally, we select $K$ context nodes where each node $j \in \mathcal{V}_{\text{ctx}}$ has the top-$K$ joint PPR score in $\hat{\boldsymbol{\pi}}(\mathcal{V}_{\text{tgt}})$. In practice, we select the context node size the same as the target node size, i.e., $K = |\mathcal{V}_{\text{tgt}}|$.

## 3.3 SPATIAL-TEMPORAL ENCODING

Given the temporal-union graph, our next step is to translate the spatial-temporal information from snapshot graphs to the temporal-union graph $\mathcal{G}_{\text{union}}$, which can be recognized and leveraged by Transformers. Notice that most classical GNNs either over-rely on the given graph structure by only considering the first- or higher-order neighbors for feature aggregation (Ying et al., 2021), or directly learn graph adjacency without using the given graph structure (Devlin et al., 2019). On the one hand, over-relying on the graph structure makes the model fails to capture the inter-relation between nodes that are not connected in the labeled graph, and could be very sensitive to the noisy edges due to human-labeling errors. On the other hand, completely ignoring the graph structure makes the optimization problem challenging because the model has to iteratively learn model parameters and estimate the graph structure. To avoid the above two extremes, we present two simple but effective designs of encodings, i.e., *temporal connection encoding* and *spatial distance encoding*, and provide details on how to integrate them into DGT.

**Temporal connection encoding.** Temporal connection (TC) encoding is designed to inform DGT if an edge $(i, j)$ exists in the $t$-th snapshot graph. We denote $\mathbf{E}^{\text{TC}} = [\mathbf{e}_{2t-1}^{\text{TC}}, \mathbf{e}_{2t}^{\text{TC}}]_{t=1}^{T} \in \mathbb{R}^{2T \times d}$ as the temporal connection encoding lookup-table where $d$ represents the hidden dimension size, which is indexed by a function $\psi(i, j, t)$ indicating whether an edge $(i, j)$ exists at time-step $t$. More specifically, we have $\psi(i, j, t) = 2t$ if $(i, j) \in \mathcal{G}_t$, $\psi(i, j, t) = 2t - 1$ if $(i, j) \notin \mathcal{G}_t$ and use this value as an index to extract the corresponding temporal connection embedding from the look-up table for next-step processing. Note that during pre-training or the training on first few time-steps, we need to mask-out certain time-steps to avoid leaking information related to the predicted items (e.g., the temporal reconstruction task in Section. 4.1). In these cases, we set $\psi(i, j, t') = \emptyset$ where $t'$ denotes the time-step we mask-out, and skip the embedding extraction at time $t'$.

**Spatial distance encoding.** Spatial distance (SD) encoding is designed to provide DGT a global view of the graph structure. The success of Transformer is largely attributed to its global receptive

field due to its full attention, i.e., each token in the sequence can attend independently to other tokens and process its representations. Computing full attention requires the model to explicitly capturing the positions dependency between tokens, which can be achieved by either assigning each position an absolute positional encoding or encode the relative distance using relative positional encoding. However, for graphs, the design of unique node positions is not mandatory because a graph is not changed by the permutation of its nodes. To encode the global structural information of a graph in the model, inspired by (Ying et al., 2021), we adopt a spatial distance encoding that measures the relative spatial relationship between any two nodes in the graph, which is a generalization of the classical Transformer's positional encoding to the graph domain. Let $D_{\max}$ be the maximum shortest path distance (SPD) we considered, where $D_{\max}$ is a hyper-parameter that can be smaller than the graph diameter. More specifically, given any node $i$ and node $j$, we define $\phi(i,j) = \min\{\text{SPD}(i,j), D_{\max}\}$ as the SPD between the two nodes if $\text{SPD}(i,j) < D_{\max}$ and otherwise as $D_{\max}$. Let $\mathbf{E}^{\text{SD}} = [\mathbf{e}_1^{\text{SD}}, \ldots, \mathbf{e}_{D_{\max}}^{\text{SD}}] \in \mathbb{R}^{D_{\max} \times d}$ as the spatial distance lookup-table which is indexed by the $\phi(i,j)$, where $\phi(i,j)$ is used to select the spatial distance encoding $\mathbf{e}_{\phi(i,j)}^{\text{SD}}$ that provides the spatial distance information of two nodes.

**Integrate spatial-temporal encoding.** We integrate temporal connection encoding and spatial distance encoding by projecting them as a bias term in the self-attention module. Specifically, to integrate the *spatial-temporal encoding* of node pair $(i,j)$ to DGT, we first gather all its associated temporal connection encodings on different time-steps as $\{\mathbf{e}_{\phi(i,j,t)}^{\text{TC}}\}_{t=1}^T$. Then, we apply weight average on all encodings over the temporal axis and projected the temporal averaged encoding as a scalar by $A_{i,j}^{\text{TC}} = Linear\big(WeightAverage(\{\mathbf{e}_{\phi(i,j,t)}^{\text{TC}}\}_{t=1}^T)\big) \in \mathbb{R}$, where the aggregation weight is learned during training. Similarly, to integrate the *spatial distance encoding*, we project the spatial distance encoding of node pair $(i,j)$ as a scalar by $A_{i,j}^{\text{SD}} = Linear(\mathbf{e}_{\phi(i,j)}^{\text{SD}}) \in \mathbb{R}$. Then, $A_{i,j}^{\text{TC}}$ and $A_{i,j}^{\text{SD}}$ are used as the bias term to the self-attention, which we describe in detail in Section 3.4.

## 3.4 GRAPH TRANSFORMER ARCHITECTURE

As shown in Figure 1, each layer in DGT consists of two towers (i.e., the target node tower and the context node tower) to encode the target nodes and the context nodes separately. The same set of parameters are shared between two towers. The two-tower structure is motivated by the fact that nodes within each group are sampled independently but there exist neighborhood relationships between inter-group nodes. Only attending inter-group nodes help DGT better capture this context information without fusing representations from irrelevant nodes. The details are as follows:

- First, we compute the self-attentions that are used to aggregate information from target nodes to context nodes (denote as "ctx") and from context nodes to target nodes (denote as "tgt"). Let define $\mathbf{H}_{\text{ctx}}^{(\ell)} \in \mathbb{R}^{|\mathcal{V}_{\text{ctx}}| \times d}$ as the $\ell$-th layer output of the context-node tower and $\mathbf{H}_{\text{tgt}}^{(\ell)} \in \mathbb{R}^{|\mathcal{V}_{\text{tgt}}| \times d}$ as the $\ell$-th layer output of the target-node tower. Then, the $\ell$th layer self-attention is computed as

$$\mathbf{A}_{\text{ctx}}^{(\ell)} = \frac{(\text{LN}(\mathbf{H}_{\text{ctx}}^{(\ell-1)})\mathbf{W}_Q^{(\ell)})(\text{LN}(\mathbf{H}_{\text{tgt}}^{(\ell-1)})\mathbf{W}_K^{(\ell)})^\top}{\sqrt{d}}, \quad \mathbf{A}_{\text{tgt}}^{(\ell)} = \frac{(\text{LN}(\mathbf{H}_{\text{tgt}}^{(\ell-1)})\mathbf{W}_Q^{(\ell)})(\text{LN}(\mathbf{H}_{\text{ctx}}^{(\ell-1)})\mathbf{W}_K^{(\ell)})^\top}{\sqrt{d}},$$

where $\text{LN}(\mathbf{H})$ stands for applying layer normalization on $\mathbf{H}$ and $\mathbf{W}_Q^{(\ell)}, \mathbf{W}_K^{(\ell)}$ are weight matrices.
- Then, we integrate spatial-temporal encoding as a bias term to self-attention as follows

$$\mathbf{P}_{\text{ctx}}^{(\ell)} = \mathbf{A}_{\text{ctx}}^{(\ell)} + \mathbf{A}_{\text{TC}}[\mathcal{V}_{\text{ctx}}; \mathcal{V}_{\text{tgt}}] + \mathbf{A}_{\text{SD}}[\mathcal{V}_{\text{ctx}}; \mathcal{V}_{\text{tgt}}], \quad \mathbf{P}_{\text{tgt}}^{(\ell)} = \mathbf{A}_{\text{tgt}}^{(\ell)} + \mathbf{A}_{\text{TC}}[\mathcal{V}_{\text{tgt}}; \mathcal{V}_{\text{ctx}}] + \mathbf{A}_{\text{SD}}[\mathcal{V}_{\text{tgt}}; \mathcal{V}_{\text{ctx}}],$$

where $\mathbf{A}_{\text{TC}}[\mathcal{V}_A; \mathcal{V}_B], \mathbf{A}_{\text{SD}}[\mathcal{V}_A; \mathcal{V}_B]$ denote the the matrix form of the projected temporal connection and spatial distance self-attention bias with row indexed by $\mathcal{V}_A$ and columns indexed by $\mathcal{V}_B$.[1]
- After that, we use the normalized $\mathbf{P}_{\text{ctx}}^{(\ell)}$ and $\mathbf{P}_{\text{tgt}}^{(\ell)}$ to propagate information between two towers, i.e.,

$$\mathbf{Z}_{\text{ctx}}^{(\ell)} = Softmax(\mathbf{P}_{\text{ctx}}^{(\ell)})\text{LN}(\mathbf{H}_{\text{tgt}}^{(\ell-1)})\mathbf{W}_V^{(\ell)} + \mathbf{H}_{\text{ctx}}^{(\ell-1)}, \quad \mathbf{Z}_{\text{tgt}}^{(\ell)} = Softmax(\mathbf{P}_{\text{tgt}}^{(\ell)})\text{LN}(\mathbf{H}_{\text{ctx}}^{(\ell-1)})\mathbf{W}_V^{(\ell)} + \mathbf{H}_{\text{tgt}}^{(\ell-1)}.$$

- Finally, a residual connected feed-forward network is applied to the aggregated message to produce the final output

$$\mathbf{H}_{\text{ctx}}^{(\ell)} = FFN(\text{LN}(\mathbf{Z}_{\text{ctx}}^{(\ell)})) + \mathbf{Z}_{\text{ctx}}^{(\ell)}, \quad \mathbf{H}_{\text{tgt}}^{(\ell)} = FFN(\text{LN}(\mathbf{Z}_{\text{tgt}}^{(\ell)})) + \mathbf{Z}_{\text{tgt}}^{(\ell)},$$

where $FFN(\cdot)$ denotes the multi-layer feed-forward network. The final layer output of the target node tower $\mathbf{H}_{\text{tgt}}^{(L)}$ will be used to compute the loss defined in Section 4.

---

[1] Given a matrix $\mathbf{A} \in \mathbb{R}^{m \times n}$, the element at the $i$-th row and $j$-th column is denoted as $A_{i,j}$, the submatrix formed from row $\mathcal{I}_{\text{row}} = \{a_1, \ldots, a_r\}$ and columns $\mathcal{I}_{\text{col}} = \{b_1, \ldots, b_s\}$ is denoted as $\mathbf{A}[\mathcal{I}_{\text{row}}; \mathcal{I}_{\text{col}}]$.

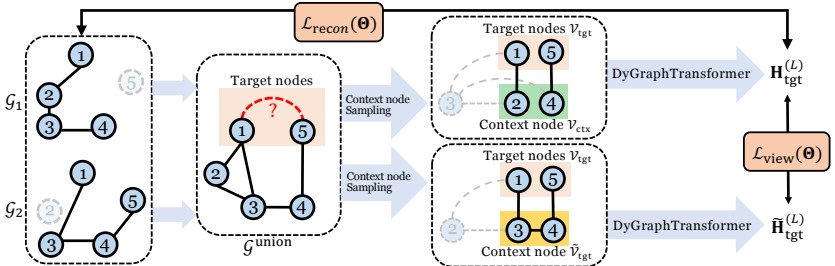

Figure 2: Overview of the pre-training. Given snapshot graphs $\{\mathcal{G}_1, \mathcal{G}_2\}$ as input, we first generate the temporal union graph. Then, we sample the target node $\mathcal{V}_{\text{tgt}}$ and two different set of context nodes $\mathcal{V}_{\text{ctx}}, \widetilde{\mathcal{V}}_{\text{ctx}}$. After that, we apply DGT on $\{\mathcal{V}_{\text{tgt}}, \mathcal{V}_{\text{ctx}}\}$ and $\{\mathcal{V}_{\text{tgt}}, \widetilde{\mathcal{V}}_{\text{ctx}}\}$ to output $\mathbf{H}_{\text{tgt}}^{(L)}$ and $\widetilde{\mathbf{H}}_{\text{tgt}}^{(L)}$. To this end, we optimize $\mathcal{L}_{\text{view}}(\boldsymbol{\Theta})$ by maximizing the similarity between $\mathbf{H}_{\text{tgt}}^{(L)}$ and $\widetilde{\mathbf{H}}_{\text{tgt}}^{(L)}$, and optimize $\mathcal{L}_{\text{recon}}(\boldsymbol{\Theta})$ by recovering snapshot graphs using $\mathbf{H}_{\text{tgt}}^{(L)}$.

## 4 DYNAMIC GRAPH TRANSFORMER LEARNING

Transformer usually requires a significant amount of supervised data to guarantee their generalization ability on unseen data. However, existing dynamic graph datasets are relatively small and may not be sufficient to train a powerful Transformer. To overcome this challenge, we propose to first pre-train DGT with two complementary self-supervised objective functions (in Section 4.1). Then, we fine-tine DGT using the supervised objective function (in Section 4.2). Notice that the same set of snapshot graphs but different objective functions are used for pre-training and fine-tuning. Finally, via an information-theoretic analysis, we show that the representation can have a better generalization ability on downstream tasks by optimizing our pre-training losses (in Section 4.3).

### 4.1 PRE-TRAINING

We introduce a *temporal reconstruction loss* $\mathcal{L}_{\text{recon}}(\boldsymbol{\Theta})$ and a *multi-view contrastive loss* $\mathcal{L}_{\text{view}}(\boldsymbol{\Theta})$ as our self-supervised object functions. Then, our overall pre-taining loss is defined as $\mathcal{L}_{\text{pre-train}}(\boldsymbol{\Theta}) = \mathcal{L}_{\text{recon}}(\boldsymbol{\Theta}) + \gamma \mathcal{L}_{\text{view}}(\boldsymbol{\Theta})$, where $\gamma$ is a hyper-parameter that balances the importance of two pre-taining tasks as illustrated in Figure 2.

**Temporal reconstruction loss.** To ensure that the spatial-temporal encoding is effective and can inform DGT the temporal dependency between multiple snapshot graphs, we introduce a temporal reconstruction loss as our first pre-training objective. Our goal is to reconstruct the $t$-th graph snapshot $\mathcal{G}_t$'s structure using all but except the $t$-th graph snapshot (i.e., $\mathbb{G} \setminus \mathcal{G}_t$). Let $\bar{\mathbf{H}}_{\text{tgt}}^{(L)}(t)$ denote the target-node tower's final layer output computed on $\mathbb{G} \setminus \mathcal{G}_t$. To decode the graph structure of graph snapshot $\mathcal{G}_t$, we use a fully connected layer as the temporal structure decoder that takes $\bar{\mathbf{H}}_{\text{tgt}}^{(L)}(t)$ as input and output $\mathbf{E}(t) = Linear(\bar{\mathbf{H}}_{\text{tgt}}^{(L)}(t)) \in \mathbb{R}^{|\mathcal{V}_{\text{tgt}}| \times d}$ with $\mathbf{e}_i(t) \in \mathbb{R}^d$ denotes the $i$-th row of $\mathbf{E}(t)$. Then, the temporal reconstruction loss is defined as $\mathcal{L}_{\text{recon}}(\boldsymbol{\Theta}) = \sum_{t=1}^{T} LinkPredLoss(\{\mathbf{e}_i(t)\}_{i \in \mathcal{V}_{\text{tgt}}}, \mathcal{V}_{\text{tgt}}, \mathcal{E}_t)$, where $\sigma(\cdot)$ is Sigmoid function and

$$LinkPredLoss(\{\mathbf{x_i}\}_{i \in \mathcal{S}}, \mathcal{S}, \mathcal{E}) \triangleq \sum_{i,j \in \mathcal{S}} \left( - \sum_{(i,j) \in \mathcal{E}} \log(\sigma(\mathbf{x}_i^\top \mathbf{x}_j)) - \sum_{(i,j) \notin \mathcal{E}} \log(1 - \sigma(\mathbf{x}_i^\top \mathbf{x}_j)) \right). \quad (1)$$

**Multi-view contrastive loss.** Recall that $\mathcal{V}_{\text{ctx}}$ is constructed by *deterministically* selecting the common neighbors of $\mathcal{V}_{\text{tgt}}$ with the top-$K$ PPR score. Then, we introduce $\widetilde{\mathcal{V}}_{\text{ctx}}$ as the subset of the common neighbors of $\mathcal{V}_{\text{tgt}}$ *randomly* sampled with sampling probability of each node proportional to its PPR score. Since a different set of context nodes are provided for the same set of target nodes, $\{\mathcal{V}_{\text{tgt}}, \widetilde{\mathcal{V}}_{\text{ctx}}\}$ provides an alternative view of $\{\mathcal{V}_{\text{tgt}}, \mathcal{V}_{\text{ctx}}\}$ when computing the representation for nodes in $\mathcal{V}_{\text{tgt}}$. Notice that although the provided context nodes are different, since they have the same target nodes, it is natural to expect the calculated representation have high similarity. We denote $\mathbf{H}_{\text{tgt}}^{(L)}$ and $\widetilde{\mathbf{H}}_{\text{tgt}}^{(L)}$ as the final layer model output that are computed on $\{\mathcal{V}_{\text{tgt}}, \mathcal{V}_{\text{ctx}}\}$ and $\{\mathcal{V}_{\text{tgt}}, \widetilde{\mathcal{V}}_{\text{ctx}}\}$. To this end, we introduce our second self-supervised objective function as $\mathcal{L}_{\text{view}}(\boldsymbol{\Theta}) = \|\mathbf{H}_{\text{tgt}}^{(L)} - StopGrad(\widetilde{\mathbf{H}}_{\text{tgt}}^{(L)})\|_{\text{F}}^2 + \|StopGrad(\mathbf{H}_{\text{tgt}}^{(L)}) - \widetilde{\mathbf{H}}_{\text{tgt}}^{(L)}\|_{\text{F}}^2$, where $StopGrad$ denotes stop gradient. Note that optimizing $\mathcal{L}_{\text{view}}(\boldsymbol{\Theta})$ alone without stopping gradient results in a degenerated solution (Chen & He, 2021; Tian et al., 2021).

## 4.2 FINE-TUNING

To apply the pre-trained model for downstream tasks, we choose to finetune the pre-trained model with downstream task objective functions. Here, we take link prediction as an example. Our goal is to predict the existence of a link at time $T + 1$ using information up to time $T$. Let $\mathbf{H}_{\text{tgt}}^{(L)}(\{\mathcal{G}_j\}_{j=1}^t)$ denote the final layer output of DGT using snapshot graphs $\{\mathcal{G}_j\}_{j=1}^t$. Then, the link prediction loss is defined as $\mathcal{L}_{\text{LinkPred}}(\mathbf{\Theta}) = \sum_{t=1}^{T-1} LinkPredLoss(\mathbf{H}_{\text{tgt}}^{(L)}(\{\mathcal{G}_j\}_{j=1}^t), \mathcal{V}_{\text{tgt}}, \mathcal{E}_{t+1})$, where $LinkPredLoss$ is defined in Eq. 1.

## 4.3 ON THE IMPORTANCE OF PRE-TRAINING FROM INFORMATION THEORY PERSPECTIVE

In this section, we show that our pre-training objectives can improve the generalization error under mild assumptions and results in a better performance on downstream tasks. Let $X$ denote the input random variable, $S$ as the self-supervised signal (also known as a different view of input $X$), and $Z_X = f(X), Z_S = f(S)$ as the representations that are generated by a deterministic mapping function $f$. In our setting, we have the sampled sub-graph of temporal-union graph $\mathcal{G}^{\text{union}}$ induced by node $\{\mathcal{V}_{\text{tgt}}, \mathcal{V}_{\text{ctx}}\}$ as input $X$, the sampled subgraph of $\mathcal{G}^{\text{union}}$ induced by node $\{\mathcal{V}_{\text{tgt}}, \mathcal{V}_{\text{ctx}}\}$ as self-supervised signal $S$, and DGT as $f$ that computes the representation of $X, S$ by $Z_X = f(X), Z_S = f(S)$. Besides, we introduce the task-relevant information as $Y$, which refers to the information that is required for downstream tasks. For example, when the downstream task is link prediction, $Y$ can be the ground truth graph structure about which we want to reason. Notice that in practice we have no access to $Y$ during pre-training and it is only introduced as the notation for analysis. Furthermore, let $H(A)$ denote entropy, $H(A|B)$ denote conditional entropy, $I(A; B)$ denote mutual information, and $I(A; B|C)$ denote conditional mutual information. More details and preliminaries on information theory are deferred to Appendix B.

In the following, we study the generalization error of the learned representation $Z_X$ under the binary classification setting. We choose *Bayes error rate* (i.e., the lowest possible test error rate a binary classifier can achieve) as our evaluation metric, which can be formally defined as $P_e = 1 - \mathbb{E}[\max_y P(Y = y|Z_X)]$. Before proceeding to our result, we make the following assumption on input $X$, self-supervised signal $S$, and task-relevant information $Y$.

**Assumption 1.** *We assume the task-relevant information is shared between the input random variable $X$, self-supervised signal $S$, i.e., we have $I(X; Y|S) = 0$ and $I(S; Y|X) = 0$.*

We argue the above assumption is mild because input $X$ and self-supervised signal $S$ are two different views of the data, therefore they are expected to contain task-relevant information $Y$. In Proposition 1, we make connections between the Bayes error rate and pre-training losses, which explains why the proposed pre-training losses are helpful for downstream tasks. We defer the proofs to Appendix B.

**Proposition 1.** *We can upper bound Bayes error rate by $P_e \leq 1 - \exp(-H(Y) + I(Z_X; X) - I(Z_X; X|Y))$, and reduce the upper bound of $P_e$ by (1) maximizing the mutual information $I(Z_X; X)$ between the learned representation $Z_X$ and input $X$, which can be achieved by minimizing temporal reconstruction loss $\mathcal{L}_{recon}(\mathbf{\Theta})$, and (2) minimizing the task-irrelevant information between the learned representation $Z_X$ and input $X$, which can be achieved by minimizing our multi-view loss $\mathcal{L}_{view}(\mathbf{\Theta})$.*

The Proposition 1 suggests that if we can create a different views $S$ of our input data $X$ in a way such that both $X$ and $S$ contain the task-relevant information $Y$, then by jointly optimizing our two pre-training losses can result in the representation $Z_X$ with a lower Bayes error rate $P_e$. Our analysis is based on the information theory framework developed in (Tsai et al., 2020), in which they show that using contrastive loss between $Z_X$ and $S$ (i.e., maximizing $I(Z_X; S)$), predicting $S$ from $Z_X$ (i.e., minimizing $H(S|Z_X)$), and predicting $Z_X$ from $S$ (i.e., minimizing $H(Z_X|S)$) can result in a smaller Bayes error rate $P_e$.

## 5 EXPERIMENTS

We evaluate DGT using dynamic graph link prediction, which has been widely used in (Sankar et al., 2018; Goyal et al., 2018) to compare its performance with a variety of static and dynamic graph representation learning baselines. Besides, DGT can also be applied to other downstream tasks such as node classification. We defer node classification results to Appendix A.3.

Table 1: Comparing DGT with baselines using *Micro-* and *Macro-AUC* on real-world datasets.

| Method | Metric | Enron | RDS | UCI | Yelp | ML-10M |
|---|---|---|---|---|---|---|
| NODE2VEC | Micro-AUC | $82.42 \pm 2.03$ | $81.10 \pm 0.87$ | $81.41 \pm 0.60$ | $68.93 \pm 0.33$ | $90.50 \pm 0.83$ |
| | Macro-AUC | $81.35 \pm 2.93$ | $82.85 \pm 0.86$ | $81.39 \pm 0.76$ | $67.38 \pm 0.49$ | $89.48 \pm 0.62$ |
| GRAPHSAGE | Micro-AUC | $82.39 \pm 3.01$ | $85.49 \pm 0.96$ | $79.85 \pm 2.62$ | $62.36 \pm 1.01$ | $86.31 \pm 0.97$ |
| | Macro-AUC | $83.41 \pm 2.94$ | $86.64 \pm 0.89$ | $78.45 \pm 2.01$ | $58.36 \pm 0.91$ | $90.23 \pm 0.90$ |
| DYNAERNN | Micro-AUC | $74.00 \pm 1.24$ | $80.56 \pm 0.77$ | $79.29 \pm 1.90$ | $71.54 \pm 0.83$ | $87.01 \pm 0.88$ |
| | Macro-AUC | $74.36 \pm 1.35$ | $80.16 \pm 0.91$ | $83.81 \pm 1.25$ | $72.29 \pm 0.58$ | $89.04 \pm 0.67$ |
| DYNGEM | Micro-AUC | $66.46 \pm 0.74$ | $79.29 \pm 1.01$ | $76.36 \pm 0.83$ | $69.43 \pm 1.09$ | $79.80 \pm 0.88$ |
| | Macro-AUC | $68.46 \pm 1.14$ | $81.94 \pm 1.97$ | $78.22 \pm 0.99$ | $69.93 \pm 0.78$ | $84.86 \pm 0.49$ |
| DYSAT | Micro-AUC | $83.81 \pm 1.55$ | $83.89 \pm 0.92$ | $83.10 \pm 0.99$ | $69.00 \pm 0.22$ | $88.91 \pm 0.87$ |
| | Macro-AUC | $83.73 \pm 1.61$ | $83.60 \pm 0.68$ | $86.32 \pm 1.46$ | $69.42 \pm 0.25$ | $90.63 \pm 0.91$ |
| EVOLVEGCN | Micro-AUC | $73.83 \pm 1.23$ | $85.35 \pm 0.87$ | $85.81 \pm 0.50$ | $68.99 \pm 0.67$ | $92.79 \pm 0.21$ |
| | Macro-AUC | $75.77 \pm 1.57$ | $86.53 \pm 0.76$ | $84.18 \pm 0.72$ | $69.41 \pm 0.26$ | $93.45 \pm 0.19$ |
| **DGT** | Micro-AUC | $\mathbf{87.32 \pm 0.87}$ | $\mathbf{88.77 \pm 0.50}$ | $\mathbf{87.91 \pm 0.32}$ | $\mathbf{73.39 \pm 0.21}$ | $\mathbf{95.30 \pm 0.36}$ |
| | Macro-AUC | $\mathbf{87.82 \pm 0.89}$ | $\mathbf{89.77 \pm 0.46}$ | $\mathbf{88.49 \pm 0.43}$ | $\mathbf{74.31 \pm 0.23}$ | $\mathbf{96.16 \pm 0.22}$ |

## 5.1 EXPERIMENT SETUP

**Datasets.** We select five real-world datasets of various sizes and types in our experiments. The detailed data statistics can be accessed at Table 11 in Appendix C.2. Graph snapshots are created by splitting the data using suitable time windows such that each snapshot has an equitable number of interactions. In each snapshot, the edge weights are determined by the number of interactions.

**Link prediction task.** To compare the performance of DGT with baselines, we follow the evaluation strategy in (Goyal et al., 2018; Zhou et al., 2018; Sankar et al., 2018) by training a logistic regression classifier taking two node embeddings as input for dynamic graph link prediction. Specifically, we learn the dynamic node representations on snapshot graphs $\{\mathcal{G}_1, \ldots, \mathcal{G}_T\}$ and evaluate DGT by predicting links at $\mathcal{G}_{T+1}$. For evaluation, we consider all links in $\mathcal{G}_{T+1}$ as positive examples and an equal number of sampled unconnected node pairs as negative examples. We split 20% of the edge examples for training the classifier, 20% of examples for hyper-parameters tuning, and the rest 60% of examples for model performance evaluation following the practice of existing studies (e.g., (Sankar et al., 2018)). We evaluate the link prediction performance using Micro- and Macro-AUC scores, where the Micro-AUC is calculated across the link instances from all the time-steps while the Macro-AUC is computed by averaging the AUC at each time-step. During inference, all nodes in the testing set (from 60% edge samples in $\mathcal{G}_{T+1}$) are selected as the target nodes. To scale the inference of the testing sets of any sizes, we compute the full attention by first splitting all self-attentions into multiple chunks then iteratively compute the self-attention in each chunk (as shown in Figure 7). Since only a fixed number of self-attention is computed at each iteration, we significantly reduce DGT 's inference memory consumption. We also repeat all experiments three times with different random seeds.

**Baselines.** We compare with several state-of-the-art methods as baselines including both static and dynamic graph learning algorithms. For static graph learning algorithms, we compare against NODE2VEC (Grover & Leskovec, 2016) and GRAPHSAGE (Hamilton et al., 2017). To make the comparison fair, we feed these static graph algorithms the same temporal-union graph used in DGT rather than any single graph snapshots. For dynamic graph learning algorithms, we compare against DYNAERNN (Goyal et al., 2020), DYNGEM (Goyal et al., 2018), DYSAT (Sankar et al., 2018), and EvolveGCN (Pareja et al., 2020). We use the official implementations for all baselines and select the best hyper-parameters for both baselines and DGT. Notice that we only compare with dynamic graph algorithms that takes a set of temporal ordered snapshot graph as input, and leave the study on other dynamic graph structure (e.g., continuous time-step algorithms (Xu et al., 2020; Rossi et al., 2020)) as a future direction. More details on experiment configurations are deferred to Appendix C and more results (Figure 4, Table 2 and 3) are deferred to Appendix A.1.

## 5.2 EXPERIMENT RESULTS

Table 1 indicates the state-of-the-art performance of our approach on link prediction tasks, where DGT achieves a consistent $1\% \sim 3\%$ Macro-AUC gain on all datasets. Besides, DGT is more stable when using different random seeds observed from a smaller standard deviation of the AUC score. Furthermore, to better understand the behaviors of different methods from a finer granularity, we compare the model performance at each time-step in Figure 4 and observe that the performance of DGT is relatively more stable than other methods over time. Besides, we additionally report the results of dynamic link prediction evaluated only on unseen links at each time-step. Here, we define unseen links as the ones that first appear at the prediction time-step but are not in the previous graph snapshots. From Table 2, we find that although all methods achieve a lower AUC score, which may

Figure 3: Comparison of the *Micro-* and *Macro-AUC* score of DGT with and without pre-training.

be due to the new link prediction is more challenging, DGT still achieves a consistent $1\% \sim 3\%$ Macro-AUC gain over baselines. Moreover, we compare the training time and memory consumption with several baselines in Table 3 and shows that DGT maintains a good scalability.

### 5.3  ABLATION STUDIES

We conduct ablation studies to further understand DGT and present the details in Appendix A.2.

**The effectiveness of pre-training.**  We compare the performance of DGT with and without pre-training. As shown in Figure 3, DGT 's performance is significantly improved if we first pre-train it with the self-supervised loss with a fine-tuning on downstream tasks. When comparing the AUC scores at each time-step, we observe that the DGT with no pre-training has a relatively lower performance but a larger variance. This may be due to the vast number of training parameters in DGT , which potentially requires more data to be trained well. The self-supervised pre-training alleviate this challenge by utilizing additional unlabeled input data.

**Comparing two-tower to single-tower architecture.**  In Table 4, we compare the performance of DGT with single- and two-tower design where a single-tower means a full-attention of over all pairs of target and context nodes. We observe that the two-tower DGT has a consistent performance gain ($0.5\%$ Micro- and Macro-AUC) over the single-tower on Yelp and ML-10M. This may be due to that the nodes within the target or context node set are sampled independently while inter-group nodes are likely to be connected. Only attending inter-group nodes helps DGT better capturing these contextual information without fusing representations from irrelevant nodes.

**Comparing $K$-hop attention with full attention.**  To better understand full-attention, we compare it with sparson and ions such as 1-hop and 3-hop attention. These variants are evaluated based on the single-tower DGT to include all node pairs into consideration. Table 5 shows the results where we observe that the full attention presents a consistent performance gain around $1\% \sim 3\%$ over the other two variants. This demonstrates the benefits of full-attention when modeling implicit edge connections in graphs with a larger receptive fields comparing to its $K$-hop counterparts.

**The effectiveness of spatial-temporal encoding.**  In Table 6, we conduct an ablation study by independently removing two encodings to validate the effectiveness of spatial-temporal encoding. We observe that even without any encoding (i.e., ignoring the spatial-temporal graph topologies), due to full attention, DGT is still very competitive comparing with the state-of-the-art baselines in Table 1. However, we also observe a $0.6\% \sim 4.6\%$ performance gain when adding the spatial connection and temporal distance encoding, which empirically shows their effectiveness.

**The effectiveness of stacking more layers.**  When stacking more layers, traditional GNNs usually suffer from the over-smoothing (Zhao & Akoglu, 2020; Yan et al., 2021) and result in a degenerated performance. We study the effect of applying more DGT layers and show results in Table 7. In contrast to previous studies, DGT has a relatively stable performance and does not suffer much from performance degradation when the number of layers increases. This is potentially due to that DGT only requires a shallow architecture since each individual layer is capable of modeling longer-range dependencies due to full attention. Besides, the self-attention mechanism can automatically attend importance neighbors, therefore alleviate the over-smoothing and bottleneck effect.

### 6  CONCLUSION

In this paper, we introduce DGT for dynamic graph representation learning, which can efficiently leverage the graph topology and capture implicit edge connections. To further improve the generalization ability, two complementary pre-training tasks are introduced. To handle large-scale dynamic graphs, a temporal-union graph structure and a target-context node sampling strategy are designed for an efficient and scalable training. Extensive experiments on real-world dynamic graphs show that DGT presents significant performance gains over several state-of-the-art baselines. Potential future directions include exploring GNNs on continuous dynamic graphs and studying its expressive power.

**Reproducibility Statement.** We summarize the hardware specification and environment used in experiments in Appendix C.1, details on all datasets in Appendix C.2, details on baseline hyper-parameters tuning in Appendix C.3, details on DGT hyper-parameters in Appendix C.4, and the proof of theory in Appendix B. We plan to release our code afterwards or being requested by the reviewers during the review phase.

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

# Appendix

## Table of Contents

# A    MORE EXPERIMENT RESULTS

## A.1    LINK PREDICTION RESULTS

In this section, we provide the remained figures and tables in Section 5.

**Comparison of AUC score at different time steps.**    In Figure 4, we compare the AUC score of DGT with baselines on Enron, UCI, Yelp, and ML-10M dataset. We can observe that DGT can consistently outperform baselines on Enron and ML-10M dataset at all time steps, but has a relatively lower AUC score at certain time steps on the UCI and Yelp dataset. Besides, the performance of DGT is relatively more stable than baselines on different time steps.

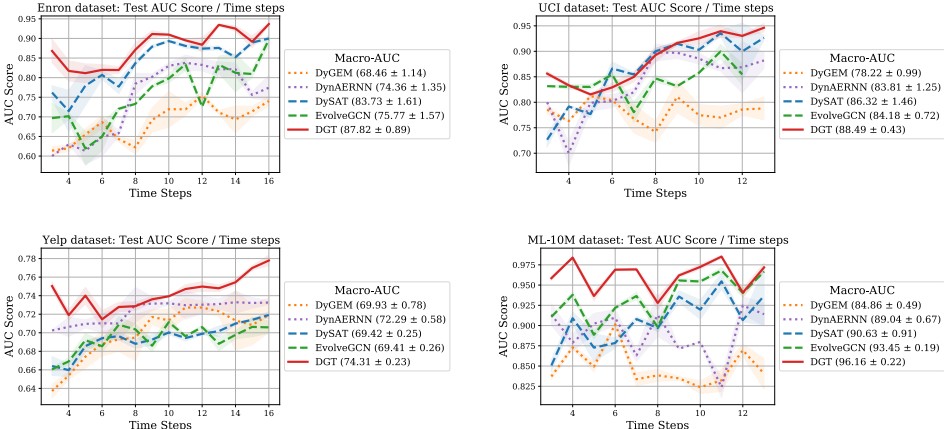

Figure 4: Comparison of DGT with baselines across multiple time steps, where the Macro-AUC score is reported in the box next to the curves.

**Comparision of AUC score on new link prediction task.**    In Table 2, we report dynamic link prediction result evaluated only on the new links at each time step, where a link that appears at the current snapshot but not in the previous snapshot is considered as a new link. This experiment can provide an in-depth analysis of the capabilities of different methods in predicting unseen links. As shown in Table 2, all methods achieve a lower AUC score, which is expected because new link prediction is more challenging. However, DGT still achieves consistent gains of $1 \sim 3\%$ Macro-AUC over baselines, thus illustrate its effectiveness in accurately temporal context for new link prediction.

Table 2: Comparison of *Micro-* and *Macro-AUC* on real-world datasets restricted to new edges.

| Method | Metric | Enron | RDS | UCI | Yelp | ML-10M |
|---|---|---|---|---|---|---|
| NODE2VEC | Micro-AUC | $76.45 \pm 1.93$ | $75.62 \pm 1.42$ | $75.31 \pm 0.83$ | $68.83 \pm 0.29$ | $88.92 \pm 0.79$ |
| | Macro-AUC | $76.01 \pm 2.39$ | $76.25 \pm 0.85$ | $75.82 \pm 0.96$ | $68.00 \pm 0.51$ | $88.01 \pm 0.50$ |
| GRAPHSAGE | Micro-AUC | $74.36 \pm 2.88$ | $80.21 \pm 0.87$ | $76.56 \pm 1.91$ | $61.97 \pm 1.00$ | $85.18 \pm 0.89$ |
| | Macro-AUC | $74.42 \pm 2.64$ | $79.99 \pm 0.78$ | $75.94 \pm 1.88$ | $58.49 \pm 0.89$ | $89.31 \pm 0.93$ |
| DYNAERNN | Micro-AUC | $61.81 \pm 1.89$ | $68.43 \pm 1.13$ | $77.39 \pm 2.10$ | $70.82 \pm 0.93$ | $86.89 \pm 0.75$ |
| | Macro-AUC | $62.62 \pm 1.91$ | $68.18 \pm 1.23$ | $81.82 \pm 1.71$ | $71.56 \pm 0.77$ | $89.45 \pm 0.53$ |
| DYNGEM | Micro-AUC | $59.42 \pm 1.42$ | $72.43 \pm 1.62$ | $74.72 \pm 0.73$ | $69.23 \pm 1.76$ | $77.18 \pm 1.96$ |
| | Macro-AUC | $61.61 \pm 1.91$ | $74.49 \pm 2.21$ | $76.34 \pm 0.78$ | $70.67 \pm 1.32$ | $82.62 \pm 0.49$ |
| DYSAT | Micro-AUC | $75.78 \pm 1.89$ | $76.28 \pm 1.34$ | $81.18 \pm 1.09$ | $69.12 \pm 0.21$ | $88.21 \pm 0.64$ |
| | Macro-AUC | $76.92 \pm 1.81$ | $76.87 \pm 1.21$ | $83.43 \pm 1.57$ | $69.20 \pm 0.20$ | $88.98 \pm 0.87$ |
| EVOLVEGCN | Micro-AUC | $67.82 \pm 1.71$ | $78.36 \pm 0.91$ | $81.99 \pm 0.73$ | $68.73 \pm 0.64$ | $90.91 \pm 0.32$ |
| | Macro-AUC | $69.39 \pm 1.89$ | $79.18 \pm 1.01$ | $82.18 \pm 0.76$ | $68.63 \pm 0.30$ | $91.45 \pm 0.29$ |
| DGT | Micro-AUC | $\mathbf{81.99 \pm 0.90}$ | $\mathbf{82.78 \pm 0.56}$ | $\mathbf{85.78 \pm 0.99}$ | $\mathbf{73.32 \pm 0.22}$ | $\mathbf{93.01 \pm 0.23}$ |
| | Macro-AUC | $\mathbf{81.32 \pm 0.89}$ | $\mathbf{82.89 \pm 0.52}$ | $\mathbf{86.21 \pm 0.56}$ | $\mathbf{73.88 \pm 0.22}$ | $\mathbf{93.56 \pm 0.21}$ |

**Computation time and memory consumption.** In Table 3, we compare the memory consumption and epoch time on the last time step of ML-10M and Yelp dataset. We chose the last time step of these two datasets because its graph size is relatively larger than others, which can provide a more accurate time and memory estimation. The memory consumption is record by `nvidia-smi` and the time is recorded by function `time.time()`. During pre-training, DGT samples 256 context node and 256 context node at each iteration. During fine-tuning, DGT first 256 positive links (links in the graph) and sample $2,560$ negative links (node pairs that do not exist in the graph), then treat all nodes in the sampled node pairs at target nodes and sample the same amount of context nodes. Notice that although the same sampling size hyper-parameter is used, since the graph size and the graph density are different, the actual memory consumption and time are also different. For example, since the Yelp dataset has more edges with more associated nodes for evaluation than ML-10M, the memory consumption and time are required on Yelp than on ML-10M dataset.

Table 3: Comparison of the epoch time and memory consumption of DGT with baseline methods on the last time step of **ML-10M** and **Yelp** dataset using the neural architecture configuration summarized in Section C.4.

| Dataset | Method | Memory consumption | Epoch time | Total time |
|---------|--------|-------------------|------------|------------|
| **ML-10M** | DYSAT | 9.2 GB | 97.2 Sec | 4276.8 Sec (45 epochs) |
| | EVOLVEGCN | 13.6 GB | 6.9 Sec | 821.1 Sec (120 epochs) |
| | DGT (Pre-training) | 6.5 GB | 38.9 Sec | 986.5 Sec (89 epochs) |
| | DGT (Fine-tuning) | 10.1 GB | 2.98 Sec | 62.2 Sec (22 epochs) |
| **Yelp** | DYSAT | 5.4 GB | 29.4 Sec | 4706.4 Sec (160 epochs) |
| | EVOLVEGCN | 7.5 GB | 19.14 Sec | 1091.2 Sec (57 epochs) |
| | DGT (Pre-training) | 21.3 GB | 11.8 Sec | 413.5 Sec (34 epochs) |
| | DGT (Fine-tuning) | 21.3 GB | 21.41 Sec | 521.6 Sec (23 epochs) |

## A.2 ABLATION STUDY RESULTS.

In this section, we provide missing the tables in Section 5.3, where discussion on the results are provided in Section 5.3.

**Compare two-tower to single-tower architecture.** In Table 4, we compare the Micro-AUC score and Macro-AUC score of DGT with one-tower[2] and two-tower structure on UCI, Yelp, and ML-10M datasets.

Table 4: Comparison of the *Micro-* and *Macro-AUC* of DGT using single-tower and two-tower model architecture on the real-world datasets.

| Method | Metric | UCI | Yelp | ML-10M |
|--------|--------|-----|------|--------|
| Single-tower | Micro-AUC | $87.86 \pm 0.60$ | $72.95 \pm 0.20$ | $94.80 \pm 0.81$ |
| | Macro-AUC | $88.27 \pm 0.68$ | $73.81 \pm 0.21$ | $95.49 \pm 0.57$ |
| Two-tower | Micro-AUC | $\mathbf{87.91 \pm 0.32}$ | $\mathbf{73.39 \pm 0.21}$ | $\mathbf{95.30 \pm 0.36}$ |
| | Macro-AUC | $\mathbf{88.49 \pm 0.43}$ | $\mathbf{74.31 \pm 0.23}$ | $\mathbf{96.16 \pm 0.22}$ |

---

[2]The node representation $\mathbf{H}^{(\ell)}$ in the single-tower DGT is computed by

$$\mathbf{H}^{(\ell)} = FFN(LN(\mathbf{Z}^{(\ell)})) + \mathbf{Z}^{(\ell)}$$

$$\mathbf{Z}^{(\ell)} = Softmax\left(\frac{(LN(\mathbf{H}^{(\ell-1)})\mathbf{W}_Q^{(\ell)})(LN(\mathbf{H}^{(\ell-1)})\mathbf{W}_K^{(\ell)})^\top}{\sqrt{d}} + \mathbf{A}_{\text{TC}} + \mathbf{A}_{\text{SD}}\right) LN(\mathbf{H}^{(\ell-1)})\mathbf{W}_V^{(\ell)} + \mathbf{H}^{(\ell-1)}.$$

$$(2)$$

**Compare $K$-hop attention with full attention.** In Table 4, we compare the performance of "single-tower DGT using full-attention", "single-tower DGT using 1-hop attention", and "single-tower DGT using 3-hop attention" on the UCI, Yelp, and ML-10M dataset.

Table 5: Comparison of the *Micro-* and *Macro-AUC* of *full attention* and $K$-*hop attention* using the single-tower architecture on the real-world datasets.

| Method | Metric | UCI | Yelp | ML-10M |
|---|---|---|---|---|
| Full attention | Micro-AUC | **87.86 ± 0.60** | **72.95 ± 0.20** | **94.80 ± 0.81** |
| | Macro-AUC | **88.27 ± 0.68** | **73.81 ± 0.21** | **95.49 ± 0.57** |
| 1-hop neighbor | Micro-AUC | 84.62 ± 0.31 | 71.33 ± 0.43 | 91.88 ± 0.73 |
| | Macro-AUC | 85.10 ± 0.15 | 71.45 ± 0.45 | 92.18 ± 0.44 |
| 3-hop neighbor | Micro-AUC | 87.01 ± 0.89 | 71.19 ± 0.22 | 91.83 ± 0.92 |
| | Macro-AUC | 87.48 ± 0.88 | 72.31 ± 0.22 | 92.33 ± 0.82 |

**The effectiveness of spatial-temporal encoding.** In Table 6, we validate the effectiveness of spatial-temporal encoding by independently removing the temporal edge coding and spatial distance encoding.

Table 6: Comparison of the *Micro-* and *Macro-AUC* of with and without spatial-temporal encoding on the real-world datasets.

| Method | Metric | UCI | Yelp | ML-10M |
|---|---|---|---|---|
| With both encoding | Micro-AUC | **87.91 ± 0.32** | **73.39 ± 0.21** | **95.30 ± 0.36** |
| | Macro-AUC | **88.49 ± 0.43** | **74.31 ± 0.23** | **96.16 ± 0.22** |
| Without any encoding | Micro-AUC | 83.27 ± 0.29 | 72.82 ± 0.37 | 91.81 ± 0.43 |
| | Macro-AUC | 83.87 ± 0.47 | 73.80 ± 0.38 | 92.59 ± 0.35 |
| Only temporal connective encoding | Micro-AUC | 84.78 ± 0.31 | 73.36 ± 0.26 | 94.51 ± 0.37 |
| | Macro-AUC | 84.60 ± 0.42 | 74.31 ± 0.25 | 95.43 ± 0.29 |
| Only spatial distance encoding | Micro-AUC | 87.01 ± 0.46 | 72.98 ± 0.32 | 92.34 ± 0.40 |
| | Macro-AUC | 87.99 ± 0.47 | 73.90 ± 0.36 | 93.13 ± 0.33 |

**The effect of the number of layers.** In Table 7, we compare the Micro-AUC score and Macro-AUC score of DGT with a different number of layers on the UCI, Yelp, and ML-10M datasets.

Table 7: Comparison of the *Micro-* and *Macro-AUC* score of DGT with different number of layers on the real-world datasets.

| Method | Metric | UCI | Yelp | ML-10M |
|---|---|---|---|---|
| 2 layers | Micro-AUC | 87.89 ± 0.43 | 74.30 ± 0.21 | 94.99 ± 0.21 |
| | Macro-AUC | 88.31 ± 0.53 | 74.29 ± 0.23 | 96.08 ± 0.15 |
| 4 layers | Micro-AUC | 87.42 ± 0.36 | **73.39 ± 0.21** | 95.30 ± 0.36 |
| | Micro-AUC | 88.35 ± 0.37 | **74.31 ± 0.23** | **96.16 ± 0.22** |
| 6 layers | Micro-AUC | **87.91 ± 0.32** | 74.30 ± 0.20 | **95.35 ± 0.28** |
| | Micro-AUC | **88.49 ± 0.43** | 74.28 ± 0.22 | 96.11 ± 0.18 |

A.3 NODE CLASSIFICATION RESULTS

In this section, we show that although DGT is orginally designed for the link prediction task, the learned representation of DGT can be also applied to binary node classification. We evaluate DGT on Wikipedia and Reddit dataset, where dataset statistic is summarized in Table 11. The snapshot is created in a similar manner as the link prediction task. As shown in Table 8 and Figure 5, DGT performs around 0.7% better than all baselines on the Wikipedia dataset and around 0.7% better than EVOLVEGCN on Reddit dataset. However, the results DGT on the Reddit dataset is slightly lower than DYSAT. This is potentially due to DGT is less in favor of a dense graph, e.g., Reddit dataset, with very dense graph structure information encoded by spatial-temporal encodings.

Table 8: Comparison of the *Micro-* and *Macro-AUC* score of DGT with different number of layers on the real-world datasets for binary node classification task.

| Method | Metric | Wikipedia | Reddit |
|--------|--------|-----------|--------|
| DYSAT | Micro-AUC | $94.69 \pm 0.46$ | $\mathbf{87.35 \pm 0.28}$ |
| | Macro-AUC | $94.74 \pm 0.66$ | $\mathbf{87.36 \pm 0.30}$ |
| EVOLVEGCN | Micro-AUC | $92.31 \pm 0.68$ | $84.72 \pm 0.89$ |
| | Micro-AUC | $92.36 \pm 0.85$ | $84.79 \pm 0.88$ |
| DGT (without pre-training) | Micro-AUC | $92.90 \pm 0.84$ | $82.37 \pm 0.78$ |
| | Micro-AUC | $92.94 \pm 0.62$ | $84.41 \pm 0.82$ |
| DGT (with pre-training) | Micro-AUC | $\mathbf{95.49 \pm 0.66}$ | $85.48 \pm 0.43$ |
| | Micro-AUC | $\mathbf{95.55 \pm 0.65}$ | $85.50 \pm 0.44$ |

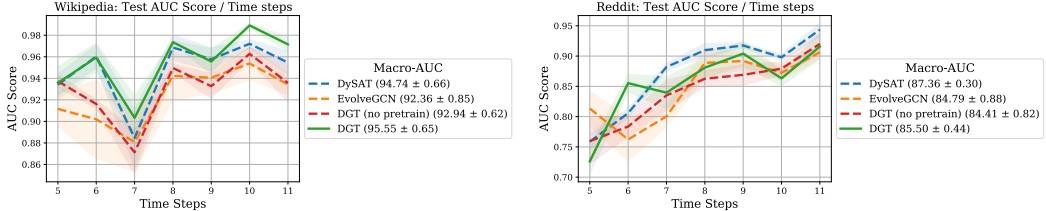

Figure 5: Comparison of DGT with baselines across multiple time steps, where the Macro-AUC score is reported in the box next to the curves

A.4 RESULTS ON NOISY DATASET

In this section, we study the effect of noisy input on the performance of DGT using *UCI* and *Yelp* datasets. We achieve this by randomly selecting 10%, 20%, 50% of the node pairs and changing their connection status either from connected to not-connected or from not-connected to connected. As shown in Table 9, although the performance of both using full-attention and 1-hop attention decreases as the noisy level increases, the performance of using full-attention aggregation is more stable and robust as the noisy level changes. This is because 1-hop attention relies more on the given structure, while full-attention only take the give structure as a reference and learns the "ground truth" underlying graph structure by gradient descent update.

Table 9: Comparison of the *Macro-AUC* score of DGT and its variants with input graph with different noisy level.

| | Method | | 10% | 20% | 50% |
|---|--------|---|-----|-----|-----|
| UCI | DGT (1-hop attention) | Micro-AUC | $82.97 \pm 0.56$ | $81.23 \pm 0.78$ | $77.85 \pm 0.66$ |
| | | Macro-AUC | $83.01 \pm 0.61$ | $82.10 \pm 0.60$ | $78.43 \pm 0.67$ |
| | DGT (Full aggregation) | Micro-AUC | $86.98 \pm 0.51$ | $86.10 \pm 0.57$ | $84.36 \pm 0.49$ |
| | | Macro-AUC | $86.12 \pm 0.57$ | $85.93 \pm 0.59$ | $85.51 \pm 0.51$ |
| Yelp | DGT (1-hop attention) | Micro-AUC | $70.00 \pm 0.20$ | $68.55 \pm 0.21$ | $65.32 \pm 0.22$ |
| | | Macro-AUC | $69.94 \pm 0.20$ | $68.45 \pm 0.23$ | $65.61 \pm 0.15$ |
| | DGT (Full aggregation) | Micro-AUC | $70.99 \pm 0.20$ | $71.74 \pm 0.19$ | $70.93 \pm 0.21$ |
| | | Macro-AUC | $71.64 \pm 0.18$ | $71.67 \pm 0.21$ | $69.93 \pm 0.21$ |

A.5 Comparison with continuous-graph learning algorithms

In this section, we compare snapshot graph-based methods against continuous graph-based learning algorithm on the *UCI*, *Yelp*, and *ML-10M* dataset. For the continuous graph learning algorithm, we choose JODIE Kumar et al. (2019) and TGAT Xu et al. (2020) as the baseline. As shwon in Table 10, JODIE and TGAT suffer from significant performance degradation. This is because they are designed to leverage the edge features and fine-grained timestamp information for link prediction, however, these information is lacking on existing snapshot graph datasets.

Please note that we compare with continuous graph algorithm only for the sake of completeness. However, since snapshot graph-based methods and continuous graph-based methods require different input graph structures, different evaluation strategies, and are designed under different settings, directly comparing two sets of methods cannot provide much meaningful interpretation. For example, existing works Kumar et al. (2019); Xu et al. (2020) on a continuous graph select the training and evaluation set by taking the first $80\%$ of links in the dataset for training and taking the rest for evaluation. In other words, the training and evaluation samples can be arbitrary close and might even come from the same time step. However, in the snapshot graph, the training and evaluation set is selected by taking the links in the previous $T-1$ snapshot graphs for training and evaluating on the $T$-th snapshot graph. That is, the training and evaluation samples never come from the same time step. Besides, since the time steps in the continuous graph are fine-grained than snapshot graphs, continuous graph methods suffer from performance degradation when applied on the snapshot graph dataset due to lack of fine-grained timestamp information. Due to the aforementioned reasons, existing continuous graph learning methods (e.g., Jodie, TGAT) only compare with other continuous graph methods on the continuous datasets, similarly, existing snapshot graph learning methods (e.g., DySAT, EvolveGCN, DynAERNN, DynGEM) also only considers other snapshot graph based methods as their baseline for a comparison.

Table 10: Comparison of the *Micro-* and *Macro-AUC* score of DGT , JODIE on the real-world datasets.

| Method | Metric | UCI | Yelp | ML-10M |
|--------|--------|-----|------|--------|
| DGT | Micro-AUC | $\mathbf{87.91 \pm 0.32}$ | $\mathbf{73.39 \pm 0.21}$ | $\mathbf{95.30 \pm 0.36}$ |
| | Macro-AUC | $\mathbf{88.49 \pm 0.43}$ | $\mathbf{74.31 \pm 0.23}$ | $\mathbf{96.16 \pm 0.22}$ |
| JODIE | Micro-AUC | $57.99 \pm 0.34$ | $59.85 \pm 0.32$ | $62.84 \pm 0.47$ |
| | Macro-AUC | $57.21 \pm 0.37$ | $61.01 \pm 0.44$ | $61.30 \pm 0.46$ |
| TGAT | Micro-AUC | $48.15 \pm 0.45$ | $51.95 \pm 0.39$ | $52.15 \pm 0.51$ |
| | Macro-AUC | $49.02 \pm 0.43$ | $52.78 \pm 0.40$ | $51.15 \pm 0.50$ |

# B  Pre-training can reduce the irreducible error

## B.1  Preliminary on information theory

In this section, we recall preliminaries on information theory, which are helpful to understand the proof in the following section. More details can be found in books such as Murphy (2022); Cover & Thomas (2006).

**Entropy.** Let $X$ be a discrete random variable, $\mathcal{X}$ as the sample space, and $x$ as outcome. We define the probability mass function as $p(x) = \Pr(X = x)$, $x \in \mathcal{X}$. Then, the *entropy* for a discrete random variable $X$ is defined as

$$H(X) = -\sum_{x \in \mathcal{X}} p(x) \log p(x), \tag{3}$$

where we use $\log$ base 2. The *joint entropy* of two random variables $X, Y$ is defined as

$$H(X, Y) = -\sum_{x \in \mathcal{X}, y \in \mathcal{Y}} p(x, y) \log p(x, y). \tag{4}$$

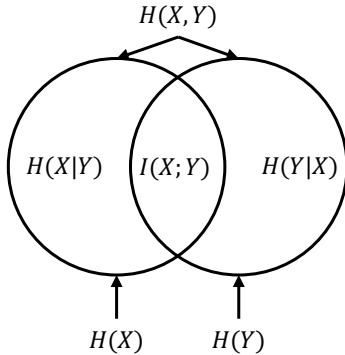

Figure 6: Relationship between entropy and mutual information (Figure 2.2 of Cover & Thomas (2006)).

The *conditional entropy* of $Y$ given $X$ is the uncertainty we have in $Y$ after seeing $X$, which is defined as

$$
\begin{aligned}
H(Y|X) &= \sum_{x \in \mathcal{X}} p(x) H(Y|X = x) \\
&= H(X,Y) - H(X).
\end{aligned}
\tag{5}
$$

Notice that we have $H(Y|X) = 0$ if $Y = f(X)$, where $f$ a deterministic mapping.

**Mutual information.** Mutual information is a special case of *KL-divergence*, which is a measure of distance between two distributions. The KL-divergence between $p(x), q(x)$ is defined as

$$
KL(p\|q) = \sum_{x \in \mathcal{X}} p(x) \log \frac{p(x)}{q(x)}.
\tag{6}
$$

Then, the mutual information $I(X;Y)$ between random variable $X, Y$ is defined as follows

$$
I(X;Y) = KL\big(p(x,y)\|p(x)p(y)\big) = \sum_{x \in \mathcal{X}, y \in \mathcal{Y}} \log \frac{p(x,y)}{p(x)p(y)},
\tag{7}
$$

and we have $I(X;Y) = 0$ if $X, Y$ are independent. Notice that we use $I(X;Y)$ instead of $I(X,Y)$ represent the mutual information between $X$ and $Y$. Besides, $I(X;Y,Z)$ represent the mutual information between $X$ and $(Y,Z)$.

Based on the above definition on entropy and mutual information, we have the following relation between entropy and mutual information:

$$
\begin{aligned}
I(X;Y) &= H(X) - H(X|Y) \\
&= H(Y) - H(Y|X) \\
&= H(X) + H(Y) - H(X,Y),
\end{aligned}
\tag{8}
$$

and the following relation between conditional entropy and conditional mutual information:

$$
\begin{aligned}
I(X;Y|Z) &= H(X|Z) - H(X|Y,Z) \\
&= H(Y|Z) - H(Y|X,Z) \\
&= H(X|Z) + H(Y|Z) - H(X,Y|Z),
\end{aligned}
\tag{9}
$$

where *conditional mutual information* $I(X;Y|Z)$ can be think of as the reduction in the uncertainty of $X$ due to knowledge of $Y$ when $Z$ is given.

A figure showing the relation between mutual information and entropy is provided in Figure 6.

**Data processing inequality.** Random variables $X, Y, Z$ are said to form a *Markov chain* $X \to Y \to Z$ if the joint probability mass function can be written as $P(x,y,z) = p(x)p(y|x)p(z|y)$. Suppose random variable $X, Y, Z$ forms a Markov chain $X \to Y \to Z$, then we have $I(X;Y) \geq I(X;Z)$.

**Bayes error and entropy.** In the binary classification setting, *Bayes error rate* is the lowest possible test error rate (i.e., irreducible error), which can be formally defined as

$$P_e = \mathbb{E}\left[1 - \max_y p(Y = y|X)\right],\tag{10}$$

where $Y$ denotes label and $X$ denotes input. Feder & Merhav (1994) derives an upper bound showing the relation between *Bayes error rate* with entropy:

$$-\log(1 - P_e) \leq H(Y|X).\tag{11}$$

The above inequality is used as the foundation of our following analysis.

### B.2 PROOF OF PROPOSITION 1

In the following, we utilize the analysis framework developed in Tsai et al. (2020) to show the importance of two pre-training loss functions.

By using Eq. 11, we have the following inequality:

$$-\log(1 - P_e) \leq H(Y|Z_X)\tag{12}$$

By rearanging the above inequality, we have the following upper bound on the Bayes error rate

$$
\begin{aligned}
P_e &\leq 1 - \frac{1}{\exp\left(H(Y|Z_X)\right)} \\
&\underset{(a)}{=} 1 - \frac{1}{\exp\left(H(Y) - I(Z_X; Y)\right)} \\
&\underset{(b)}{=} 1 - \frac{1}{\exp\left(H(Y) - I(Z_X; X) + I(Z_X; X|Y)\right)},
\end{aligned}\tag{13}
$$

where equality $(a)$ is due to $I(Z_X; Y) = H(Y) - H(Y|Z_X)$, equality $(b)$ is due to $I(Z_X; Y) = I(Z_X; X) - I(Z_X; X|Y) + I(Z_X; Y|X)$ and $I(Z_X; Y|X) = 0$ because $Z_X = f(X)$ is a deterministic mapping given input $X$. Our goal is to find the deterministic mapping function $f$ to generate $Z_X$ that can maximize $I(Z_X; X) - I(Z_X; X|Y)$, such that the upper bound on the right hand side of Eq. 13 is minimized. We can achieve this by:

- Maximizing the mutual information $I(Z_X; X)$ between the representation $Z_X$ to the input $X$.
- Minimizing the task-irrelevant information $I(Z_X; X|Y)$, i.e., the mutual information between the representation $Z_X$ to the input $X$ given task-relevant information $Y$.

In the following, we first show that minimizing $\mathcal{L}_{\text{recon}}(\Theta)$ can maximize the mutual information $I(Z_X; X)$, then we show that minimizing $\mathcal{L}_{\text{view}}(\Theta)$ can minimize the task irrelevant information $I(Z_X; X|Y)$.

**Maximize mutual information $I(Z_X; X)$.** By the relation between mutual information and entropy $I(Z_X; X) = H(X) - H(X|Z_X)$, we know that maximizing the mutual information $I(Z_X; X)$ is equivalent to minimizing the conditional entropy $H(X|Z_X)$. Notice that we ignore $H(X)$ because it is only dependent on the raw feature and is irrelevant to feature representation $Z_X$. By the definition of conditional entropy, we have

$$
\begin{aligned}
H(X|Z_X) &= \sum_{z_x \in \mathcal{Z}_\mathcal{X}} p(z_x) H(X|Z_X = z_x) \\
&= \sum_{z_x \in \mathcal{Z}_\mathcal{X}} p(z_x) \sum_{x \in \mathcal{X}} -p(x|z_x) \log p(x|z_x) \\
&= \sum_{z_x \in \mathcal{Z}_\mathcal{X}} \sum_{x \in \mathcal{X}} -p(x, z_x) \log p(x|z_x) \\
&= \mathbb{E}_{\mathrm{P}(X, Z_X)}\left[-\log \mathrm{P}(X|Z_X)\right] \\
&= \min_{Q_\theta} \mathbb{E}_{\mathrm{P}(X, Z_X)}\left[-\log Q_\theta(X|Z_X)\right] - \mathrm{KL}\left(\mathrm{P}(X|Z_X)\|Q_\theta(X|Z_X)\right) \\
&\leq \min_{Q_\theta} \mathbb{E}_{\mathrm{P}(X, Z_X)}\left[-\log Q_\theta(X|Z_X)\right]
\end{aligned}\tag{14}
$$

where $Q_{\boldsymbol{\theta}}(\cdot|\cdot)$ is a variational distribution with $\boldsymbol{\theta}$ represent the parameters in $Q_{\boldsymbol{\theta}}$ and KL denotes KL-divergence.

Therefore, maximizing mutual information $I(Z_X; X)$ can be achieved by minimizing $\mathbb{E}_{\mathrm{P}_{X,Z_X}}[-\log Q_{\boldsymbol{\theta}}(X|Z_X)]$. By assuming $Q_{\boldsymbol{\theta}}$ as the categorical distribution and $\boldsymbol{\theta}$ as a neural network, minimizing $\mathbb{E}_{\mathrm{P}_{X,Z_X}}[-\log Q_{\boldsymbol{\theta}}(X|Z_X)]$ can be think of as introducing a neural network parameterized by $\boldsymbol{\theta}$ to predict the input $X$ from the learned representation $Z_X$ by minimizing the binary cross entropy loss.

**Minimize the task irrelevant information** $I(Z_X; X|Y)$**.** Recall that in our setting, input $X$ is the node features of $\{\mathcal{V}_{\text{target}}, \mathcal{V}_{\text{context}}\}$ and the subgraph induced by $\{\mathcal{V}_{\text{target}}, \mathcal{V}_{\text{context}}\}$. The self-supervised signal $S$ is node features of $\{\mathcal{V}_{\text{target}}, \widetilde{\mathcal{V}}_{\text{context}}\}$ and the subgraph induced by $\{\mathcal{V}_{\text{target}}, \widetilde{\mathcal{V}}_{\text{context}}\}$. Therefore, it is natural to make the following mild assumption on the input random variable $X$, self-supervised signal $S$, and task relevant information $Y$.

**Assumption 2.** *We assume tall task-relevant information is shared between the input random variable* $X$*, self-supervised signal* $S$*, i.e., we have* $I(X; Y|S) = 0$ *and* $I(S; Y|X) = 0$*.*

In the following, we show that minimizing $I(Z_X; X|Y)$ can be achieved by minimizing $H(Z_X|S)$. From data processing inequality, we have $I(X; Y|S) \geq I(Z_X; Y|S) \geq 0$. From Assumption 2, we have $I(X; Y|S) = 0$, therefore we know $I(Z_X; Y|S) = 0$. By the relation between mutual information and entropy, we have

$$
\begin{aligned}
I(Z_X; X|Y) &= H(Z_X|Y) - H(Z_X|X, Y) \\
&\underset{(a)}{=} H(Z_X|Y) \\
&= H(Z_X|S, Y) + I(Z_X; S|Y) \\
&= H(Z_X|S) - I(Z_X; Y|S) + I(Z_X; S|Y) \\
&\underset{(b)}{=} H(Z_X|S) + I(Z_X; S|Y) \\
&\underset{(c)}{\leq} H(Z_X|S) + I(X; S|Y),
\end{aligned}
\tag{15}
$$

where equality $(a)$ is due to $H(Z_X|X, Y) = 0$ since $Z_X = f(X)$ and $f$ is a deterministic mapping, equality $(b)$ is due to $I(Z_X, Y|S) = 0$, and inequality $(c)$ is due to data processing inequality.

From Eq. 14, we know that

$$
\begin{aligned}
H(Z_X|S) &= \mathbb{E}_{\mathrm{P}(S, Z_X)}[-\log \mathrm{P}(Z_X|S)] \\
&\leq \min_{Q'_{\phi}} \mathbb{E}_{\mathrm{P}(S, Z_X)}\Big[-\log Q'_{\phi}(Z_X|S)\Big].
\end{aligned}
\tag{16}
$$

By assuming $Q'_{\phi}$ as the Gaussian distribution and $\phi$ as a neural network, minimizing $\mathbb{E}_{\mathrm{P}_{S,Z_X}}[-\log Q_{\phi}(Z_X|S)]$ can be think of as introducing a neural network parameterized by $\phi$ that take $S$ as input and output $Z_S = \text{NeuralNetwork}_{\phi}(S)$, then minimize the mean-square error between $Z_X$ and $Z_S$.

## C EXPERIMENT CONFIGURATION

### C.1 HARDWARE SPECIFICATION AND ENVIRONMENT

We run our experiments on a single machine with Intel i9-10850K, Nvidia RTX 3090 GPU, and 32GB RAM memory. The code is written in Python 3.7 and we use PyTorch 1.4 on CUDA 10.1 to train the model on the GPU.

### C.2 DETAILS ON DATASETS

We summarize the dataset statistic in Table 11. More specifically, we prepare snapshot graphs following the procedure as described in Sankar et al. (2018). In the following, we provide brief descriptions of each dataset.

Table 11: Statistics of the datasets used in our experiments.

|  | **Enron** | **DRS** | **UCI** | **Yelp** | **ML-10M** | **Wikipedia** | **Reddit** |
|---|---|---|---|---|---|---|---|
| **Nodes** | 143 | 167 | 1,809 | 6,569 | 20,537 | 9,227 | 11,000 |
| **Edges** | 2,347 | 1,521 | 16,822 | 95,361 | 43,760 | 157,474 | 672,447 |
| **Time steps** | 16 | 100 | 13 | 16 | 13 | 11 | 11 |

- **Enron dataset**[3]: Enron is a public available social network dataset which contains data from about 150 users. We only consider the email communications between Enron employees to generate the dynamic dataset, and use a 2 month sliding window to construct 16 snapshots.
- **RDS dataset**[4]: This is a publicly available social network dataset that contains an email communication network between employees of a mid-sized manufacturing company Radoslaw. Nodes represent employees and edges represent individual emails between two users. The snapshots are created using a window size of 3 days.
- **UCI dataset**[5]: This is a publicly available social network dataset that contains private messages sent between users on an online social network platform at the University of California, Irvine over 6 months. The snapshots are created using a window size of 10 days.
- **Yelp dataset**[6]: This is a public available rating dataset which contains user-business rating in Arizona. We only consider user-business pairs that have at least 15 interactions. The snapshots are created using a window size of 6 months.
- **ML-10M**[7]: This is a publicly available rating dataset that contains tagging behavior of MovieLens users, with the tags applied by a user on her rated movies. The snapshots are created using a window size of 3 months.
- **Wikipedia**[8]: This is a publicly available interaction graph, where users and pages are nodes, and an interaction represents a user editing a page. The snapshots are created using a window size of 3 days.
- **Reddit**[9]: This is a publicly available interaction graph, where users and subreddits are nodes, and interaction occurs when a user writes a post to the subreddit. The snapshots are created using a window size of 3 days. The data is from a pre-existing, publicly available dataset collected by a third party.

## C.3 DETAILS ON BASELINE HYPTER-PARAMETERS TUNING

We tune the hyper-parameters of baselines following their recommended guidelines.

- NODE2VEC[10]: We use the default setting as introduced in Grover & Leskovec (2016). More specifically, for each node we use 10 random walks of length 80, context window size as 10. The in-out hyper-parameter $p$ and return hyper-parameter $q$ are selected by grid-search in range $\{0.25, 0.5, 1, 2, 5\}$ on the validation set.
- GRAPHSAGE[11]: We use the default setting as introduced in Hamilton et al. (2017). More specifically, we train two layer GNN with neighbor sampling size 25 and 10. The neighbor aggregation is selected by grid-search from "mean-based aggregation", "LSTM-based aggregation", "max-pooling aggregation", and "GCN-based aggregation" on the validation set. In practice, GCN aggregator performs best on Enron, RDS, and UCI, and max-pooling aggregator performs best on Yelp and ML-10M.

---

[3]https://www.cs.cmu.edu/~enron/
[4]https://nrvis.com/download/data/dynamic/ia-radoslaw-email.zip
[5]http://konect.cc/networks/opsahl-ucsocial/
[6]https://www.yelp.com/dataset
[7]https://grouplens.org/datasets/movielens/10m/
[8]http://snap.stanford.edu/jodie/wikipedia.csv
[9]http://snap.stanford.edu/jodie/reddit.csv
[10]https://github.com/aditya-grover/node2vec
[11]https://github.com/williamleif/GraphSAGE

- DynGEM and DynAERNN[12]: We use the default setting as introduced in Goyal et al. (2018) and Goyal et al. (2020). The scaling and regularization hyper-parameters is selected by grid-search in range $\alpha \in \{10^{-6}, 10^{-5}\}$, $\beta \in \{0.1, 1, 2, 5\}$, and $\nu_1, \nu_2 \in \{10^{-6}, 10^{-4}\}$ on the validation set.
- DySAT[13]: We use the default setting and model architecture as introduced in Sankar et al. (2018). The co-occurring positive node pairs are sampled by running 10 random walks of length 40 for each node. The negative sampling ratio is selected by grid-search in the range $\{0.01, 0.1, 1\}$, number of the self-attention head is selected in the range $\{8, 16\}$, and the feature dimension is selected in the range $\{128, 256\}$ on the validation set.
- EvolveGCN[14]: We use the default setting and model architecture as introduced in Pareja et al. (2020). We train both EvolveGCN-O and EvolveGCN-H and report the architecture with the best performance on the validation set. In practice, EvolveGCN-O performs best on UCI, Yelp, and ML-10M, EvolveGCN-H performs best on Enron and RDS.

## C.4 Additional details on experiment configuration

We select a single set of hyper-parameters for each dataset using grid search. More specifically, we select the negative sampling ratio (i.e., number of positive edge/number of negative edge) in the range $\{0.01, 0.1, 1\}$, number of the self-attention head is selected in the range $\{8, 16\}$, feature dimension is selected in the range $\{128, 256\}$, number of layers in the range $\{2, 4, 6\}$, maximum shortest path distance $D_{\max}$ in the range $\{2, 3, 5\}$ on the validation set, and the hyper-parameter that balances the importance of two pre-taining tasks $\gamma = 1$. The hyper-parameter for each dataset is summarized in Table 12.

Table 12: Hyper-parameters used in DGT for different datasets. "-" stands for hyper-parameters that are not required.

|  | Enron | RDS | UCI | Yelp | ML-10M | Wiki | Reddit |
|---|---|---|---|---|---|---|---|
| Self-attention head | 16 | 16 | 8 | 8 | 8 | 8 | 8 |
| Hidden dimension | 256 | 256 | 128 | 128 | 128 | 128 | 128 |
| Number of layer | 2 | 2 | 6 | 4 | 4 | 2 | 2 |
| Hidden feature dropout out ratio | 0.5 | 0.5 | 0.5 | 0.5 | 0.5 | 0.5 | 0.5 |
| Self-attention dropout ratio | 0.1 | 0.1 | 0.1 | 0.5 | 0.5 | 0.1 | 0.1 |
| Negative sampling ratio | 50 | 10 | 10 | 10 | 10 | — | — |
| Maximize shortest path distance | 5 | 5 | 5 | 5 | 5 | 5 | 5 |
| Mini-batch size | 512 | 512 | 512 | 512 | 512 | 800 | 800 |

During training, we set the target node size as the largest number our GPU (Nvidia RTX 3090) can fit, and sample the context nodes the same size as target nodes. In practice, we found that a mini-batch size of 512 works well on all datasets. During the evaluation, we set all validation and testing nodes as target nodes during evaluation. Notice that since gradients are not required during evaluation, therefore we can obtain all the self-attention values by first splitting all self-attentions into multiple chunks, then we iterative compute the self-attention in each chunk on GPU as shown in Figure 7. Since every iteration only a fixed number of self-attention are computed and computing the self-attention is the most memory-consuming operation, DGT can inference validation and testing set of any size.

## D Comparison of different graph Transformer

In this section, we provide details on the comparison of different graph Transformers as summarized in Table 13, provide detailed information on different positional encoding type in Section D.1, and discussion on othe application of graph Transformers in Section D.2.

---

[12]https://github.com/palash1992/DynamicGEM
[13]https://github.com/aravindsankar28/DySAT
[14]https://github.com/IBM/EvolveGCN

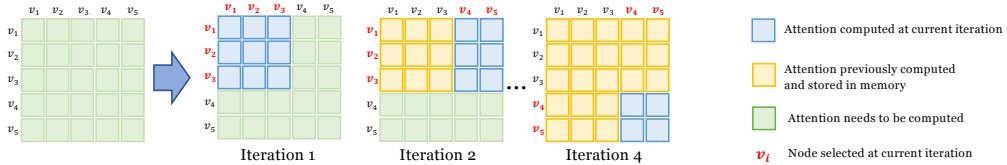

Figure 7: Suppose we want to compute the pairwise self-attention between nodes $\{v_1, \ldots, v_5\}$. We can first split all self-attentions into multiple chunks, and iteratively compute the self-attention value in each chunk. For example, at the first iteration, we first compute the self attention between node $\{v_1, \ldots, v_3\}$ (in blue) and store it in memory (in yellow). Then, at the second iteration, we compute the attention between node $\{v_1, \ldots, v_3\}$ and node $\{v_4, v_5\}$.

Table 13: Comparison of different graph Transformers.

| Method | GRAPHORMER | GRAPHTRANSFORMER | GRAPHBERT | DGT (ours) |
|---|---|---|---|---|
| Mini-batch | No | No | Yes | Yes |
| Graph type | Static | Static | Static | Dynamic |
| Attention type | Full-attention | 1-hop attention | Full-attention | Full-attention |
| Encoding type | Centrality Encoding Spatial Encoding Edge Encoding | Laplacian Eigenvector | WL-based Intimacy-based Hop-based | Temporal Connection Spatial Distance |

### D.1 POSITIONAL ENCODING TYPES

In the following, we summarize the positional encoding of different methods.

(1) GRAPHORMER (Ying et al., 2021):

- Centrality Encoding: the in-degree and out-degree information of each node. Then, they add the degree information to the original node feature.
- Spatial Encoding: shortest path distance between two nodes. Then, they add the shortest path distance as a bias term to the self-attention.
- Edge Encoding: the summation of all egde features on the shortest path. Then, they add the shortest path distance as a bias term to the self-attention.

(2) GRAPHTRANSFORMER (Dwivedi & Bresson, 2020) proposes to use the Laplacian Eigenvectors. Then, they add the Eigenvectors to the original node feature.

(3) GRAPHBERT (Zhang et al., 2020):

- Weisfeiler-Lehman Absolute Role Embedding: the node label generated by the Weisfeiler-Lehman (WL) algorithm according to the structural roles of each node in the graph data. Then, they add the labelled node information to the original node feature.
- Intimacy based Relative Positional Embedding: the placement orders of the serialized node list ordered by the Personalized PageRank score of each node. Then, they add the node order information to the original node feature.
- Hop based Relative Distance Embedding: shortest path distance between a node to the center node of the sampled subgraph. Then, they add the shortest path distance as a bias information to the original node feature.

### D.2 GRAPH TRANSFORMER WITH OTHER APPLICATIONS

Transformers are also applied to other graph types or downstream tasks. For example, HGT Hu et al. (2020) and GTNs Yun et al. (2019) propose a first-order graph transformer to solve the heterogeneous graph representation learning problem, TAGGEN Zhou et al. (2020) is working on synthetic graph generation problems, which are out of the scope of this paper.

