# OpenReview forum: "Dynamic Graph Representation Learning via Graph Transformer Networks"
_ICLR.cc/2022/Conference — ICLR 2022 Submitted_

### Official Review · Reviewer_8DU1 · 2021-10-23

**Correctness:** 3
**Technical Novelty And Significance:** 2
**Empirical Novelty And Significance:** 2
**Recommendation:** 3
**Confidence:** 4

**Main Review:**

This paper makes an attempt to study the dynamic graph representation learning. The paper proposes a couple tricks for the combination with graph transformer networks. The tricks include sampling, union graph, pretraining etc. Some experiments have been conducted.

This paper makes an attempt to study the dynamic graph representation learning. The paper proposes a couple tricks for the combination with graph transformer networks. The tricks include sampling, union graph, pretraining etc. Some experiments have been conducted.

While the topic of the paper: time series modeling of graphs is certainly important and needs to be highlighted, i find the paper doesn't really touch upon the core research question of what models are good for time series modeling of graphs, i.e., how the time domain and cross-section domain should be best combined. Instead, the paper mostly focused on a few engineering tricks for applying the transformer architecture and some pretraining tasks. This makes the paper quite disappointing and not so important for the research community, hence, I recommend rejection. This paper is maybe more suited for engineering or Kaggle style conferences, but not ICLR, which should highlight eye-opening scientific research.

1. While the tricks you propose for Transformer is of interest to some Kaggle audience, it's not clear what's the best way to combine time and cross-section domain information. The simplest you can do is either treat each time snapshot as independent samples, and do basic training, this in practice work quite well. You can also use say RNN or TCN etc but unclear whether these will add value. It's also not clear what's the best ways. In this sense, this paper doesn't really touch upon this question at all. Authors need to work on this.

2. Authors need to implement and compare with the efficient Transformer models such as Liformer etc. It's not clear the tricks proopsed can outperform those.

3. If the graphs are quite noisy, say the financial time series, then will attention fail?Since the main focus is on attention, authors need to validate this.




**Summary Of The Paper:**

This paper makes an attempt to study the dynamic graph representation learning. The paper proposes a couple tricks for the combination with graph transformer networks. The tricks include sampling, union graph, pretraining etc. Some experiments have been conducted.

**Summary Of The Review:**

While the topic of the paper: time series modeling of graphs is certainly important and needs to be highlighted, i find the paper doesn't really touch upon the core research question of what models are good for time series modeling of graphs, i.e., how the time domain and cross-section domain should be best combined. Instead, the paper mostly focused on a few engineering tricks for applying the transformer architecture and some pretraining tasks. This makes the paper quite disappointing and not so important for the research community, hence, I recommend rejection. This paper is maybe more suited for engineering or Kaggle style conferences, but not ICLR, which should highlight eye-opening scientific research.

---

> ### Author Response · Authors · 2021-11-18
> **Response to Reviewer 8DU1**
>
>
> > **Q1: This paper proposes a couple of engineering tricks, is quite disappointing, not important to the community, more suitable for kaggle.**
>
> A1.
> We believe this submission focuses on solving fundamental problems on dynamic graph learning using Graph Transformer, more than just simple engineering tricks.
> Specifically, to the best of our knowledge, we first propose an efficient transformer-based modeling architecture that can best utilizes both the spatial and temporal information, exactly as the review states, is critical in dynamic graph representation learning.
> Through the proposed two-tower transformer architecture with spatial-temporal encoding, DGT consistently demonstrates a significant performance gain over several state-of-the-art dynamic graph representation learning methods.
> In addition, we further analyze how pre-training can facilitate optimizing the proposed module from both empirical observation and theoretical proofs.
>
> Many existing research shares a similar research paradigm we applied: e.g., AlexNet [1] propose parallel tower network structures to optimize memory cost; BERT [2] analyzes how pre-training tasks can benefit model optimization; GraphSAGE [3] demonstrates how simple techniques like sampling can yield substantial impact on graph learning and guides future research directions in this area. We believe the techniques proposed by these work are solid and fundamental research contribution more than "a couple of enginering tricks".
>
> In summary, we think our proposed methods are novel research contribution, which improves the state-of-the-art methods in dynamic graph representation learning and helps the community better understands how transformers can be applied on dynamic graphs.
>
> [1] [ImageNet Classification with Deep Convolutional Neural Networks](https://papers.nips.cc/paper/2012/hash/c399862d3b9d6b76c8436e924a68c45b-Abstract.html)
>
> [2] [BERT: Pre-training of Deep Bidirectional Transformers for Language Understanding](https://arxiv.org/abs/1810.04805)
>
> [3] [Inductive Representation Learning on Large Graphs](https://arxiv.org/abs/1706.02216)
>
> > **Q2: If the graphs are quite noisy, will attention fail?**
>
> A2: We study the effect of noisy input on the performance of DGT using *UCI* and *Yelp* datasets. We achieve this by randomly selecting $10\\%$, $20\\%$, $50\\%$ of the node pairs and changing their connection status either from connected to not-connected or from not-connected to connected.
> As shown in Table 9, although the performance of both using full-attention and 1-hop attention decreases as the noisy level increases, the performance of using full-attention aggregation is more stable and robust as the noisy level changes.
> This is because 1-hop attention replies more on the given structure, while full attention only takes the given structure as a reference and learns the ``ground truth'' underlying graph structure by gradient descent update.
>
> > **Q3: Compare with Linformer.**
>
> A3: Linformer is designed for neural language processing, which has a different input data structure as a graph. Besides, its model complexity is $O(n)$ with $n$ as the graph size, which cannot scale on a real-world graph. Instead, we design a sampling strategy to decouple the dependency of the computation cost on the graph size.

---

### Official Review · Reviewer_G1X6 · 2021-11-02

**Correctness:** 4
**Technical Novelty And Significance:** 3
**Empirical Novelty And Significance:** 3
**Recommendation:** 6
**Confidence:** 3

**Main Review:**

Strengths:
1. The temporal-union graph can significantly reduce the computational costs and memory usage of transformers.
2. To further reduce the computation on the spatial dimension, a new sampling strategy divides the sampled nodes into two groups called target nodes and context nodes.
3. The spatial-temporal encoding layer describes a way to encode the spatial and temporal information of a node into the network.
4. The pre-training method containing a temporal reconstruction loss and a multi-view contrastive loss is proven to be useful to enhance the generalization ability of the model from both theoretical and experimental ways.
5. Experiment results on two datasets in different applications show the proposed model achieves state-of-the-art performances. Meanwhile, the ablation studies are sufficient to evaluate modules in the DGT.

Weaknesses:
1. The paper claims its main contributions to the transformers from both its title and introduction section. However, the main body of the transformer in Section 3.4 seems similar to the previous works. Contributions (2) and (3) could be considered as auxiliary methods to improve the transformer. This might be solved by a better paper organization.
2. An issue of temporal-union graph generation: the method only assumes the dynamic edges but the number of nodes remains static. This may limit the application of the model. Can the temporal-union generation be applied to the above case?
3. In Section 3.2, this paper divides the nodes into target and context nodes. However, during testing, if all the nodes in the dataset are required to be classified or to predict links (i.e., all nodes are target nodes), it seems that this sampling method and the two-tower transformer will become invalid.
4. For spatial-temporal encoding, the method is effective but a little complex. It would be better for authors to compare the performance with other straightforward approaches. For example, concatenating the spatial and temporal information to the node, using the raw information as the bias term, or other methods the authors mention in Appendix D.1.


**Summary Of The Paper:**

In this study, the authors propose a new graph transformer network for dynamic graph representation. To solve the challenges of static graphs learning and the temporal information aggregating, this paper introduces a Dynamic Graph Transformer (DGT) which contains three components: (1) a two-tower Transformer-based method, (2) temporal-union graph construction (3) a complementary pre-training task. Extensive experiments on the two datasets of link prediction and node classification demonstrate the superiority of the model. The ablation studies justify the effectiveness of each module in the DGT model.

**Summary Of The Review:**

This work proposes a new DGT for dynamic graph representation. All the claims of existing challenges and their corresponding solutions are correct. The novelty of this work is moderate and the experiments on two datasets justify the superiority of the model.

---

> ### Author Response · Authors · 2021-11-18
> **Response to Reviewer G1X6**
>
>
> > **Q1: the main body of the transformer in Section 3.4 seems similar to the previous works. Contributions (2) and (3) could be considered as auxiliary methods to improve the transformer.**
>
> A1.
> The difference between existing Graph Transformer algorithms mainly narrows down to how the information is encoded and how the self-attention is computed. We summarized the key difference between existing graph transformers in Appendix D.
>
> Besides, we think
> contribution 2 (i.e., temporal-union graph data structure and target-context node sampling strategy) and contribution 3 (i.e., pre-training tasks and our theoretical analysis on the importance of pre-training) are more than just auxiliary methods to improve the transformer.
> First of all, the design of *temporal-union graph data structure* and *target-context node sampling strategy* is to resolve computation issues in applying full attention for graph representation learning. Without them, existing work can only consider using 1-hop neighbor for feature aggregation (similar to GAT's aggregation) due to the high computation cost of full self-attention.
> Furthermore, our pre-training task is designed in a way that can have a theoretical guarantee on its beneficial to the downstream task.
> The analysis is novel and interesting because such kind of theoretical understanding (although empirically we know pre-training is beneficial) is still lacking in the existing graph pre-training works.
>
> > **Q2: the method only assumes the dynamic edges but the number of nodes remains static.**
>
> A2:
>  Our method supports dynamic nodes. Let $\mathcal{V}_t$ as the node-set for only appear at the $t$-th snapshot graph. Then, the shared node-set $\mathcal{V} = \text{unique}(\\{\mathcal{V}_t\\}_\{t=1}^T)$ is the set of node we are using to compute the node representations. When a new snapshot graph arrives, we just need to update $\mathcal{V}$ to include the new nodes. Please note that our setting is the same as other snapshot-based dynamic learning algorithms, e.g., DySAT and EvolveGCN.
> We clarify this in Section 2.
>
> > **Q3: However, during testing, if all the nodes in the dataset are required to be classified or to predict links, it seems that this sampling method and the two-tower transformer will become invalid.**
>
> A3: Indeed, during the evaluation phase, all nodes can be used for node representation (without sampling). However, we have to emphasize that the design of the sampling method and the two-tower transformer are used to reduce the computation cost during training and to provide a better node representation. As we have shown in our ablation study, using a two-tower structure gives us very stable improvement over one-tower structure in Table 4 (Appendix A.2)
>
> > **Q4: For spatial-temporal encoding, the method is effective but a little complex. It would be better for authors to compare the performance with other straightforward approaches.**
>
> A4: We evaluate the effectiveness of spatial-temporal encoding using ablation study, i.e., every time remove one component to see its effect on model performance. However, we would also love to test more simple encoding methods as a future direction.

---

### Official Review · Reviewer_uub9 · 2021-11-02

**Correctness:** 3
**Technical Novelty And Significance:** 3
**Empirical Novelty And Significance:** 2
**Recommendation:** 5
**Confidence:** 4

**Main Review:**

Strength:
The paper proposed a temporal-spatial encoding to capture implicit edge connections.
Weaknesses:
1. This paper investigates the methods for dynamic graph representation learning. However, some closely related methods are not discussed and not compared in this paper, e.g.,
https://storage.googleapis.com/pub-tools-public-publication-data/pdf/19698cbb225b3fc165231dd2fc6f0483ff28a777.pdf,
https://snap.stanford.edu/jodie/
etc.
The authors are suggested to compare with or at least discuss these work, which could provide a more comprehensive discussion for the paper.

2. The paper lacks some standard datasets that are widely utilized in this topic, e.g., MOOC and LastFM (https://snap.stanford.edu/jodie/). The authors are suggested to utilize or at least discuss these datasets, which could help the readers to compare the proposed method with the existing ones.

**Summary Of The Paper:**

In this paper, the author focused on the problem of dynamic graph representation learning and proposed a temporal-union graph structure and a target-context node sampling strategy. Experiments show that this work performs well in some real-world datasets.

**Summary Of The Review:**

In this paper, the author introduced a DGT for dynamic graph representation learning to leverage the graph topology and capture implicit edge connections. For the weaknesses, it is suggested to provide more concrete experimental results to better validate the effectiveness of the proposed method.

---

> ### Author Response · Authors · 2021-11-18
> **Responce to Reviewer uub9**
>
>
> > **Q1&Q2: Missing related works: DyREP, Jodie; missing dataset: MOOC and LastFM used in Jodie**
>
> A1&A2: Since our method is a snapshot graph-based method, but DyREP and Jodie are continuous graphs learning-based methods, we cannot directly compare the snapshot graph-based methods with the continuous graph learning-based methods or run snapshot graph-based methods on the continuous graph datasets.
>
> To provide evidence on the above statement, we compare snapshot graph-based methods against continuous graph-based learning algorithm on the UCI, Yelp, and ML-10M dataset. We change our snapshot graph dataset into continuous graph format by assigning each node a timestamp as the order of the snapshot graph (i.e., if a node $v_i$ appears at the $t$-th snapshot graph, then the timestamp $t$ is assigned to that node). For the continuous graph learning algorithm, we choose [JODIE](https://github.com/srijankr/jodie) and [TGAT](https://github.com/StatsDLMathsRecomSys/Inductive-representation-learning-on-temporal-graphs) as the baseline.
> We do not use DyREF as suggested by the reviewer because we could not find its official implementation and TGAT is the more recent state-of-the-art continuous graph-based method.
> As shown in Table 10, JODIE and TGAT suffer from significant performance degradation issues. This is because these methods are designed to leverage the edge features and fine-grained timestamp information for link prediction, however, that information is lacking on existing snapshot graph datasets.
> Please refer to Appendix A.4 for more details.

---

### Official Review · Reviewer_SMmY · 2021-11-02

**Correctness:** 3
**Technical Novelty And Significance:** 3
**Empirical Novelty And Significance:** 2
**Recommendation:** 5
**Confidence:** 4

**Main Review:**

Strength:
The proposed method was presented in detail, with many experiments to support the main contribution.

Weakness:
1. The difference among the three categories classified in "Dynamic graph representation learning" could be stated. For example, the second and third classes, according to the authors, only differ from the choice of graph convolution: "...use self-attention mechanism for spatial and temporal message aggregation. For example, DYSAT (Sankar et al., 2018) replaces the GCN with GAT...". Rather than the key difference of embedding a dynamic graph, such difference choices are more about neighborhood aggregation (no matter the graph is static or dynamic).
2. Some sections could be reallocated to make it easier to follow. For example, Section 3.1 could be moved to later sections after introducing basic paradigms, such as " temporal-union graph".
3. It is confusing whether the authors work on static nodes (and dynamic edges) or dynamic nodes. Section 2 defines "...the t-th snapshot graph Gt(V, Et) is an undirected graph with a shared node set V..." while Figure 1 (and later presentations) seems to allow different node sets at different time steps.
4. As the proposed method is claimed to be robust to noises of input data, it would be desired to design experiments to prove this statement.
5. The scalability of the proposed method could be supported by larger datasets.
6. As the model complexity relies largely on edge (according to Section 3.1), revealing the consumption on the two (relatively) large datasets, Wikipedia and Reddit, would be more informative (Table 3).

**Summary Of The Paper:**

The authors designed a graph transformer for dynamic graph link prediction tasks. The model provides a scalable solution to graph transformers, and is claimed to be robust to noise.

**Summary Of The Review:**

The paper could be more persuasive on their main contributions for providing more empirical evidence, such as the 'robustness to noise' and 'scalable solution'.

---

> ### Author Response · Authors · 2021-11-18
> **Responce to Reviewer SMmY**
>
> > **Q1: Suggestions on representations: (Related work section) In related work, the author summarized the difference between recurrent-based methods and the attention-based methods is their spatial aggregation rather than the temporal aggregation. (Method section) Some sections could be reallocated to make it easier to follow. For example, move section 3.1 after introducing temporal-union graphs.**
>
> A1:
> (**Related work section**) In the related works, we also discussed the temporal difference. For example, we highlight that
>
> - DySAT replaces RNN with self-attention for temporal aggregation;
> - TGAT directly encodes the information to the node features rather than using RNN.
>
> We've revised the draft to make this section clearer.
>
> (**Method section**) We organize our section by making sure the previous section can provide enough background on the following sections. For example, Section 3.1 introduces
> the *computation challenges* and *basic concepts on temporal union graph*, where (1) the *computation challenges* are used to motivate our sampling  in Section 3.2 and two-tower structure in Section 3.4; (2) the *basic concepts on temporal union graph* are used to introduce spatial-temporal encoding in Section 3.3 and our graph transformer architecture in Section 3.4.
>
> We will improve the presentation to make it more clear and can reflect the logic better.
>
> > **Q2: It is confusing whether the authors work on static nodes (and dynamic edges) or dynamic nodes.**
>
> A2:
>  Our method supports dynamic nodes, where nodes can be added or deleted overtime. Please note that our setting is the same as other snapshot-based dynamic learning algorithms, e.g., DySAT and EvolveGCN.
>
>  Formally, let $\mathcal{V}_t$ as the node-set for only appear at the $t$-th snapshot graph. Then, the shared node-set $\mathcal{V}$ is the set of nodes we are using to compute the node representations. When a new snapshot graph arrives, we just need to update $\mathcal{V}=\text{unique}(\\{\mathcal{V}_t\\}\_{t=1}^T)$ to include the new nodes.
>  We clarify this in Section 2.
>
> > **Q3: Experimentally show DGT is robust to noises input**
>
> A3: As suggested by the reviewer, we further study the effect of noisy input on the performance of DGT using *UCI* and *Yelp* datasets. We achieve this by randomly selecting $10\\%$, $20\\%$, $50\\%$ of the node pairs and changing their connection status either from connected to not-connected or from not-connected to connected.
>
> As shown in Table 9, although the performance of both using full-attention and 1-hop attention decreases as the noisy level increases, the performance of using full-attention aggregation is more stable and robust as the noisy level changes.
> This is because 1-hop attention relies more on the given structure, while full attention only takes the given structure as a reference and learns the ``ground truth'' underlying graph structure by gradient descent updates.
>
> > **Q4: Experiment on larger dataset**
>
> A4: The datasets used in our paper (e.g., ML-10M and Yelp) are already the biggest datasets used in snapshot graph-based dynamic graph learning problems and are used in the most recent snapshot graph-based dynamic graph learning papers, such as DySAT and EvolveGCN.
>
> Although the dataset [Elliptic](https://www.kaggle.com/ellipticco/elliptic-data-set) used in EvolveGCN is bigger than the largest dataset ML-10M we used, the nodes (in Elliptic dataset) of different time steps are disconnected, therefore not suitable for link prediction. In EvolveGCN, they use the Elliptic dataset for node-level tasks.

---

### Author Response · Authors · 2021-11-18
**Response to all reviewers**

We appreciate all constructive comments and suggestions from all reviewers.
We have revised the draft based on the reviews with the following changes highlighted in red.

- In Section 2, we clarify the definition of dynamic graphs and highlight that DGT supports adding and removing nodes for each input graph snapshot.
- In Appendix A.4, we conduct additional experiments to evaluate the robustness of DGT on noisy graphs and present the results.
- In Appendix A.5, we compare dynamic graph representation learning methods on discrete dynamic graphs with the techniques on continuous dynamic graphs with additional experiment results.

---

### Decision · Program_Chairs · 2022-01-20

**Decision:**

Reject

**Comment:**

In this study, the authors propose a new graph transformer network for dynamic graph representation. To solve the challenges of static graphs learning and the temporal information aggregating, this paper introduces a Dynamic Graph Transformer (DGT) which contains three components: (1) a two-tower Transformer-based method, (2) temporal-union graph construction (3) a complementary pre-training task. Extensive experiments on the two datasets of link prediction and node classification demonstrate the superiority of the model. The ablation studies justify the effectiveness of each module in the DGT model. The reviewers has various technical issues with the paper which the authors mostly addressed (e.g., whether the nodes are static or dynamic, whether DGT is robust to noise, whether it scales to larger datasets). Overall, the contributions seem incremental. There is confusion among the reviewers as to whether the proposed model differs from prior art. It seems to me there actual  differences  but whether they major or minor is open to interpretation. Overall there reviewers were not particularly excited about model/results/contributions.